# Decoupling Skeleton and Flesh: Efficient Multimodal Table Reasoning with Disentangled Alignment and Structure-aware Guidance

Yingjie Zhu [1 2]  Xuefeng Bai [1]  Kehai Chen [1]  Yang Xiang [2]  Youcheng Pan [2]  Xiaoqiang Zhou [2]  Min Zhang [1]

## Abstract

Reasoning over table images remains challenging for Large Vision-Language Models (LVLMs) due to complex layouts and tightly coupled structure–content information. Existing solutions often depend on expensive supervised training, reinforcement learning, or external tools, limiting efficiency and scalability. This work addresses a key question: *how to adapt LVLMs to table reasoning with minimal annotation and no external tools?* Specifically, we first introduce DISCO, a Disentangled Structure–Content alignment framework that explicitly separates structural abstraction from semantic grounding during multimodal alignment, efficiently adapting LVLMs to tables structures. Building on DISCO, we further present Table-GLS, a Global-to-Local Structure-guided reasoning framework that performs table reasoning via structured exploration and evidence-grounded inference. Extensive experiments across diverse benchmarks demonstrate that our framework efficiently enhances LVLM's table understanding and reasoning capabilities, particularly generalizing to unseen table structures. Our data and code are available at https://github.com/AAndy-Zhu/TableVLM.

## 1. Introduction

Tables are structured data representations that systematically organize information into rows and columns, serving as a fundamental medium for conveying relational data across numerous domains such as financial reports, scientific articles, medical records and government documents (Chen et al., 2021; Akhtar et al., 2022; Cheng et al., 2022; Si et al., 2023; Li et al., 2024). Recent advances in

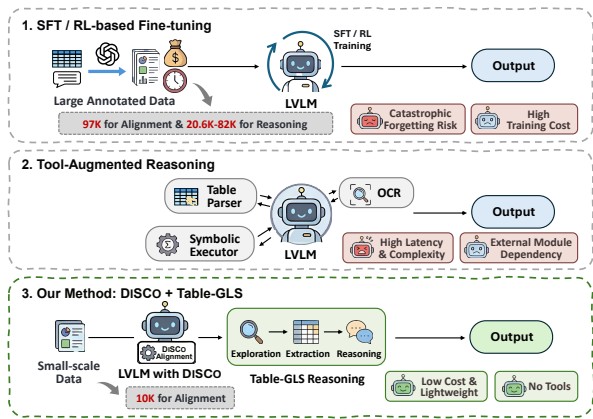

Figure 1. **Comparison of our framework with current methods.**

large foundation models have provided new modeling paradigms for automated table understanding. In particular, Large Vision-Language Models (LVLMs) (Liu et al., 2024a; Bai et al., 2025a) integrate visual perception with language modeling (Ruan et al., 2025), enabling unified processing of table images and scanned documents for real-world applications, and providing a unified framework for interpreting table images and answering natural language questions about their content (Zheng et al., 2024; Fu et al., 2025). Despite their success on various vision-language tasks (Bai et al., 2025b; Zhang et al., 2025c; Zhu et al., 2025; Wang et al., 2025; Huang et al., 2026b; Xu et al., 2026), LVLMs still struggle with table understanding and reasoning—particularly when tables exhibit complex layouts, dense data, or intricate structural dependencies.

Current methods for adapting LVLMs to table reasoning largely follow two paradigms. The first relies on extensive supervised fine-tuning (Zheng et al., 2024; Zhao et al., 2024; Zhou et al., 2025) or reinforcement learning-based optimization (Kang et al., 2025; Jiang et al., 2025) to equip models with table reasoning capabilities. While often effective, this paradigm is constrained by the need for costly and scarce expert annotations for table reasoning data, which limits scalability and risks catastrophic forgetting of the model's original reasoning skills. The second category augments LVLMs with external tools—such as visual editors or symbolic modules—to explicitly steer visual attention and reasoning steps (Fu et al., 2025). Nevertheless,

[1]Harbin Institute of Technology, Shenzhen, China [2]Peng Cheng Laboratory, Shenzhen, China. Correspondence to: Xuefeng Bai <baixuefeng@hit.edu.cn>, Yang Xiang <xiangy@pcl.ac.cn>.

*Proceedings of the 43rd International Conference on Machine Learning*, Seoul, South Korea. PMLR 306, 2026. Copyright 2026 by the author(s).

these methods increase system complexity and inference latency, failing to enhance the model's intrinsic capacity for structural understanding and reasoning. These limitations motivate a key research question:

*Is there a way to adapt LVLMs to table reasoning with minimal annotation cost and without external tools?*

In this paper, we address this question by introducing an efficient framework that adapts LVLMs to table reasoning without expensive reasoning-specific annotations or auxiliary tools, as shown in Figure 1. The key idea is to transfer the intrinsic textual-semantic reasoning capability of LVLMs to table structure through two explicit design principles: decoupling structural perception from semantic grounding, and performing structure-aware reasoning via a global-to-local chain. Specifically, we first propose a **Di**sentangled **S**tructure–**Co**ntent (**DiSCo**) alignment framework that explicitly separates structure learning from content grounding during multimodal alignment. Concretely, our approach decomposes multimodal table understanding into complementary alignment objectives, i.e., structure alignment that learns table layouts independent of cell content, and semantic alignment that grounds table content through global and local natural language descriptions. This disentanglement not only facilitates the transfer of the LVLM's existing knowledge to the table content but also enables more data-efficient and targeted adaptation to table structures. Building upon the enhanced representations learned by DiSCo, we further introduce **Table-GLS**, a **G**lobal-to-**L**ocal **S**tructure-guided reasoning method that performs multimodal table reasoning in a step-by-step yet lightweight manner without additional fine-tuning or external tools. Instead of directly predicting answers, Table-GLS guides the LVLM to first explore the global table structure and identify task-relevant row and column indices, then extract a minimal sub-table as verifiable evidence. The final reasoning is performed based on the extracted sub-table, reducing spurious correlations and improving robustness.

Extensive experiments on 21 table understanding and reasoning tasks demonstrate that the proposed methods achieve improvements on both table understanding and reasoning, using only 10K table images for alignment. DiSCo effectively enhances structure- and content-aware understanding, while Table-GLS effectively guides LVLMs for reliable table reasoning. The combined framework delivers further performance gains on challenging multimodal table tasks, particularly for tables with unseen table structures. Our contributions can be summarized as follows:

- We propose DiSCo, a disentangled structure–content alignment framework that improves LVLMs' understanding of table structure and content, especially for complex and unseen table layouts.

- We introduce Table-GLS, a global-to-local structure-aware inference framework that enables accurate table reasoning at inference time without external tools or additional reasoning-oriented fine-tuning.

- The proposed methods achieve robust improvements in understanding and reasoning across 21 tasks and benchmarks, particularly on unseen table structures.

## 2. Related Work

### 2.1. Table Modeling

Table modeling based on language models has been widely studied, with a predominant focus on table serialization, which converts 2D tables into linear text sequences. Representative methods include row-wise serialization (e.g., TaPas (Herzig et al., 2020), TUTA (Wang et al., 2021)), structure-aware formats with special tokens (e.g., TaPEx(Liu et al., 2022)), and hybrid row–column schemes (e.g., TABBIE (Iida et al., 2021)). With the rise of LLMs, recent studies explore applying general-purpose language models (e.g., LLaMA, Gemma) to tabular tasks by first converting tables into textual formats such as CSV, JSON, Markdown, or HTML (Borisov et al., 2022). While effective, these approaches encode table structure and semantic content in an entangled manner and often suffer from scalability issues under long-context modeling (Sui et al., 2024). For LVLMs, existing methods typically rely on supervised fine-tuning or reinforcement learning (e.g., GRPO) to align table images with textual representations such as HTML, Markdown, or LaTeX, implicitly coupling structure and content during training (Zheng et al., 2024; Zhou et al., 2025; Kang et al., 2025; Jiang et al., 2025). In contrast, our work introduces a disentangled structure–content representation that enables more controllable and generalizable table understanding and reasoning in multimodal settings.

### 2.2. Multimodal Table Understanding and Reasoning

Recent research on multimodal table understanding and reasoning primarily focuses on dataset construction, unified multimodal modeling, and enhanced reasoning supervision across diverse table-centric tasks. Representative benchmarks and pretraining corpora, such as MMTab (Zheng et al., 2024), TabPedia (Zhao et al., 2024), and SynTab (Zhou et al., 2025), support diverse table perception and reasoning tasks, with models like Table-LLaVA (Zheng et al., 2024) strengthening joint visual–tabular representations via cell-level alignment. Beyond generic tables, works including Multimodal ArXiv (Li et al., 2024) and MMT-BENCH (Titiya et al., 2025) extend multimodal table reasoning to scientific documents and complex real-world scenarios by modeling fine-grained interactions among

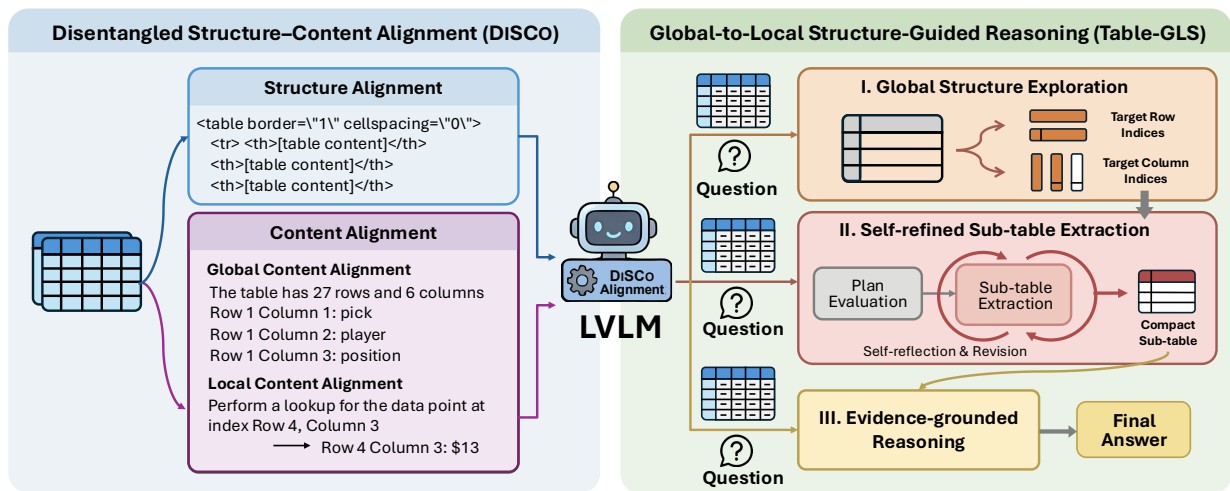

*Figure 2.* **Overall framework of DISCO and Table-GLS.**

tables, charts, text, and other visual elements. To improve reasoning ability, recent methods introduce self-training or reinforcement learning signals (Zhang et al., 2025a; Huang et al., 2026a), as exemplified by R3V (Cheng et al., 2025), Table-R1 (Kang et al., 2025), and TURBO (Jiang et al., 2025), which leverage reasoning trajectories or structure-aware rewards to optimize multimodal table reasoning during training. In contrast, inference-time approaches, such as REFOCUS (Fu et al., 2025), enable explicit multi-hop visual reasoning via external visual editing and code-driven control. Unlike current methods, which either rely on heavy training supervision or introduce additional tools, our work enhances LVLMs' table understanding and reasoning by strengthening their intrinsic structure–content modeling and enabling lightweight, structure-guided reasoning.

## 3. Methodology

We propose an efficient and unified framework for multimodal table understanding and reasoning that strengthens LVLMs at both the representation and inference levels. The **core idea** is that disentangling structure abstraction from semantic grounding during multimodal alignment allows LVLMs to transfer their inherent understanding and reasoning ability to table images in a data-efficient manner. Thus, as illustrated in Figure 2, we first introduce DISCO (§3.1), which performs disentangled structure–content alignment to enhance table structure learning by explicitly decoupling structure abstraction from semantic grounding during multimodal alignment. Then, we propose Table-GLS (§3.2), a global-to-local, structure-guided reasoning framework that performs inference by progressively identifying relevant table regions and reasoning over compact sub-tables. Together, DISCO and Table-GLS form an efficient approach for adapting LVLMs to multimodal table reasoning.

### 3.1. Disentangled Structure–Content Alignment

Tables naturally combine **structural organization** (e.g., rows, columns, and cell layout) with **semantic content** (e.g., the textual values within cells) (Lu et al., 2025). However, existing LVLM-based table alignment methods typically align table images with *linearized textual representations*, such as HTML, Markdown, or LaTeX. While these formats preserve both structure and content, they inevitably entangle layout cues with cell semantics into a single sequence, forcing the model to learn structural relations and semantic meanings simultaneously during alignment. This tightly coupled supervision obscures the distinct roles of structure and content, leading to inefficient learning and poor transfer across table layouts. Thus, we propose DISCO, which disentangles structure abstraction from semantic grounding during alignment, enabling LVLMs to leverage their existing semantic understanding ability and adapt to table structures with minimal additional supervision.

**Structure Alignment.** To explicitly model table structure, we perform structure alignment using supervision that focuses solely on layout and relational organization, independent of cell semantics. Specifically, We derive a structure representation $T_S$ by anonymizing all cell contents in conventional textual table sequences $T$, i.e. HTML, Markdown and LaTeX, using a unified placeholder token $t_p$, preserving only layout-related tokens such as row and column delimiters, hierarchical headers, and span markers:

$$T_S = \texttt{Anonymize}(T, t_p) \qquad (1)$$

Given an instruction $I_S$ and a table image $V$, the goal of structure alignment is to train the LVLMs to predict $T_S$ conditioned on the visual input, without access to semantic cell values. The training objective is formulated as

$$\mathcal{L}_{\text{struct}} = -\mathbb{E}_{(I_S, V, T)} \log P_\theta(T_S \mid I_S, V). \qquad (2)$$

This objective forces the LVLM to focus on extracting and organizing structure information directly from the table image, rather than memorizing or entangling semantic content. As a result, the learned representations capture table layout more explicitly, providing a robust structure foundation for subsequent content grounding and reasoning.

**Content Alignment.** While structure understanding provides the foundation for table reasoning, accurate interpretation of semantic content is equally critical. Therefore, we design new content alignment objectives that explicitly condition semantic prediction on global and local structure context. Specifically, at the global level, the LVLM is instructed to produce a lightweight **semi-structured description** $T_G$ of the table, including the total number of rows and columns, followed by a concise summary of what each row and column contains.

$$\mathcal{L}_{\text{content\_global}} = -\mathbb{E}_{(I_G, V, T_G)} \log P_\theta(T_G \mid I_G, V), \quad (3)$$

where $I_G$ and $V$ represent the input instruction and the table image, respectively. This task encourages the model to associate semantic meanings with structural axes, rather than individual cells in isolation.

At the local level, we further introduce targeted content querying. Given a specified row index $m$ and column index $n$, the LVLM is trained to identify and describe the textual content associated with that structural unit, such as *Row m Column n: [content]*. Thus the training objective for local content alignment is defined as

$$\mathcal{L}_{\text{content\_local}} = -\mathbb{E}_{(I_L, V, m, n, T_L)} \log P_\theta(T_L \mid I_L, V, m, n), \quad (4)$$

where $T_L$ denotes the textual content corresponding to the queried row and column. By separating content alignment from structure abstraction, this design compels the LVLM to ground semantics onto explicit structural coordinates learned during structure alignment. Together, global and local content alignment enable LVLMs to form disentangled yet complementary representations of table structure and content, crucial for robust multimodal table understanding and reasoning. More details are description in Appendix B.

### 3.2. Global-to-Local Structure-Guided Reasoning

Building upon DISCO, we introduce a Global-to-Local Structure-Guided Reasoning (Table-GLS) framework to guide LVLMs progressively reason on table structures. Different from exist methods which require extensive training or external tools, Table-GLS operates in a training-free and tool-free manner. As shown in Figure, Table-GLS guides the reasoning process through a three-stage mechanism, i.e., (I) Global Structure Exploration, (II) Self-refined Sub-table Extraction and (III) Evidence-grounded Reasoning.

**Global Structure Exploration.** Table-GLS starts with global structure exploration. Given a table image $V$ and a question $q$, the LVLM is prompted with instruction $I_{GSE}$ to analyze the overall table layout, including headers, row labels, and their semantic roles, and to determine the structural regions that are most relevant to the task,

$$T_t, R, C = \text{LVLM}(I_{GSE}, V, q), \quad (5)$$

where $T_t$ denotes a brief reasoning process that explains why certain rows or columns are needed, $R$ and $C$ are the lists of target column headers and row labels, respectively. This formulation encourages the model to reason at the level of table structure, rather than directly accessing cell-level content, enforcing a deliberate *where-to-look* decision before any content is extracted.

**Self-refined Sub-table Extraction.** The second step performs structure-guided sub-table extraction with self-reflective verification. Instead of directly extracting content from the predicted structural indices, the LVLM is first prompted to assess whether the target rows $R$ and columns $C$ obtained in the previous stage are correct and sufficient for answering the question. Specifically, given the table image $V$, the question $q$, and an initial reasoning plan $\{T_t, R, C\}$, the LVLM is instructed with $I_{SSE}$ to explicitly evaluate the adequacy of the plan and revise it if necessary at first, and then extract a minimal sub-table $T_{sub}$ with semi-structured description that contains only the information required to solve the task.

$$T_{sub} = \text{LVLM}(I_{SSE}, \{T_t, R, C\}, V, q). \quad (6)$$

By incorporating self-reflective verification, this step prevents error propagation from imperfect global exploration and enforces a plan-before-extract discipline. The resulting sub-table serves as compact, verifiable evidence, forming a reliable bridge between structural reasoning and final answer generation.

**Evidence-grounded Reasoning.** Finally, evidence-grounded reasoning is performed to produce the final answer based on explicit and textual visual evidence. Given the extracted sub-table $T_{sub}$, together with the original table image $V$ and the question $q$, the LVLM is required to reason over the sub-table as verifiable evidence and generate the final prediction,

$$\hat{y} = \text{LVLM}(I_{EGR}, T_{sub}, V, q), \quad (7)$$

where $\hat{y}$ denotes the predicted answer and $I_{EGR}$ is the corresponding instruction. Rather than reasoning directly over the entire table image, this formulation explicitly constrains the reasoning process to the extracted sub-table, which contains only task-relevant rows and columns. The

original table image $V$ is retained as auxiliary context to preserve visual grounding, while the sub-table $T_{sub}$ serves as the primary source of factual evidence.

By grounding reasoning on explicitly extracted evidence, our Table-GLS enforces a clear separation between *evidence selection* and *answer derivation*, reducing spurious correlations and discouraging reliance on global pattern matching. Consequently, the resulting reasoning process is more interpretable and robust for multimodal table understanding. The prompts are available in Appendix C.

## 4. Experimental Setup

### 4.1. Datasets

For table understanding, we first randomly sample 10K table images from various datasets within the pre-training corpus released by Zheng et al. (2024). For each image, we construct paired structure-alignment and content-alignment instances following the proposed DISCO framework. Then we conduct comprehensive evaluations on the table understanding tasks in MMTab (Zheng et al., 2024), which cover a broad range of table layouts and semantic querying settings, including table size dection (TSD), table cell extraction (TCE), table cell location (TCL), Merged Cell Dection (MCD) and Row&Column Extraction (RCE).

For table reasoning, we focus on representative structure-aware reasoning tasks within MMTab, including five table question answering benchmarks, i.e., WTQ (Pasupat & Liang, 2015), HiTab (Cheng et al., 2022), TAT-QA (Zhu et al., 2021), AIT-QA (Katsis et al., 2022), TabMCQ (Jauhar et al., 2016), and three table fact verification benchmarks, i.e, TabFact (Chen et al., 2020), InfoTabs (Gupta et al., 2020), and PubHealthTab (Akhtar et al., 2022). More details are presented in Appendix D.

### 4.2. Baselines

For table understanding, we compare DISCO with conventional multimodal alignment strategies that align table images with serialized **textual** representations, including HTML, Markdown, and LaTeX. To ensure a fair comparison, we report results under two settings, (I) *Textual (10K)*: models aligned using the same 10K table images as DISCO, and (II) *Textual (97K)*: models trained with the full pre-training data provided by (Zheng et al., 2024), consisting of 97K table images with 150K image-text pairs. We also report the results of TableLlama and Table-LLaVA provided by Zheng et al. (2024), which are fine-tuned on extensive table-based tasks. For table reasoning, we compare Table-GLS against *open-source LVLMs* without additional fine-tuning, including direct answering (DA) before and after DISCO alignment. We further report the results from *optimized LVLM*, including Table-LLaVA (Zheng et al.,

2024), Table-R1 (Kang et al., 2025), and HIPPO (Liu et al., 2025)[1]. In addition, we include GPT-4o-mini and GPT-5.4-mini as a *closed-source LVLM* baseline for reference.

### 4.3. Implementation Details

We fine-tune four representative LVLMs for table alignment, including Gemma3-12B (Team et al., 2025), Gemma3n-E4B-it, LLaVA-v1.6-7B (Liu et al., 2024b), and Qwen3-VL-8B-Instruct (Bai et al., 2025a), with LoRA (Hu et al., 2022) to preserve their original performance and mitigate catastrophic forgetting. For table reasoning, we evalute Gemma3n-E4B and Qwen3-VL-8B-Instruct with the vLLM (Kwon et al., 2023) framework to ensure efficient inference. All experiments are conducted in a zero-shot setting adhere to Zheng et al. (2024). More details about training and evaluation are shown in Appendix E.

## 5. Results and Analysis

### 5.1. Main Results

**Table Understanding.** Table 1 summarizes the results on multimodal table understanding tasks. Our analysis reveals several key findings: (I) DISCO consistently enhances table understanding across all evaluated LVLMs and tasks, demonstrating its generalizability beyond specific model architectures or training objectives. (II) The improvements are particularly evident on structure-sensitive tasks such as TSD, TCL, RCE and TCE, suggesting that DISCO enhances the LVLMs' ability to capture fine-grained row–column semantics. (III) Meanwhile, DISCO exhibits stronger robustness under OOD settings, especially for size detection and cell extraction tasks, highlighting its ability to generalize to unseen table layouts and distributions. (IV) Compared with *Textual-All*, which leverages the full 150K alignment samples from MMTab, DISCO achieves comparable or superior performance across most tasks with only 10K table images. This suggests that alignment quality, driven by explicit separation of structure and semantics, is more crucial than data quantity for multimodal table understanding. (V) DISCO yields consistent improvements across models of different scales. Notably, the larger models, e.g., Qwen3-VL-32B, benefit more substantially, likely because higher-capacity LVLMs can better internalize and exploit disentangled structure and semantic signals when the alignment objectives are explicitly separated. (VI) As shown in Table 14, on Qwen3-VL-8B, DISCO consistently outperforms textual alignment across all data scales from 5K to 20K images, demonstrating the advantage of our

---

[1]The results of optimized LVLM are sourced from their original papers. For HIPPO, we additionally conduct evaluations on AIT-QA, TabMCQ, and PubHealthTab using the official released model and scripts.

*Table 1.* **Results on multimodal table understanding tasks.** # $TI_A$ indicates the number of table images used for alignment. *None* denotes the original model, *Textual (10K)* and *Textual (97K)* denote the multimodal alignment with textual representations, where the former includes **the same 10K table image** as in our DISCO and the latter encompasses all alignment data provided by MMTab. *w/o $T_L$* indicates the removal of the local content alignment. More results are presented in Table 14.

| Models | Alignment (# $TI_A$) | TSD Row | TSD Column | TCL | RCE Row | RCE Column | MCD | TCE | OOD TSD Row | OOD TSD Column | OOD TCE | OOD TCL | OOD RCE Row | OOD RCE Column |
|---|---|---|---|---|---|---|---|---|---|---|---|---|---|---|
| **TableLlama+Oracle** | **None** | 5.30 | 4.40 | 0.82 | 4.34 | 5.26 | - | 9.35 | - | - | - | - | - | - |
| **TableLlama+OCR** | **None** | 3.90 | 3.70 | 0.65 | 2.82 | 2.39 | - | 3.95 | - | - | - | - | - | - |
| **Table-LLaVA 7B** | **Textual (97K)** | 33.10 | **33.20** | 29.31 | **31.43** | 37.93 | **17.14** | 19.45 | 25.20 | **16.40** | 11.28 | 26.10 | **21.97** | 18.14 |
| **Table-LLaVA 13B** | **Textual (97K)** | **34.40** | 27.60 | **29.68** | 31.07 | **41.49** | 16.52 | **19.53** | **31.60** | 14.80 | **11.38** | **26.17** | 21.94 | **18.67** |
| **Gemma3n-E4B** | None | 5.50 | 15.00 | **8.50** | 24.82 | **32.44** | 1.11 | 9.02 | 6.80 | 18.40 | 9.00 | 10.99 | 19.38 | 29.77 |
| | Textual (10K) | 9.50 | 19.70 | 8.30 | 24.54 | 23.25 | 0.71 | 8.64 | 10.80 | 23.60 | 10.20 | **12.32** | 24.20 | **37.06** |
| | DISCO *w/o $T_L$* (10K) | 9.90 | **20.20** | 4.71 | 27.45 | 23.69 | 0.76 | 11.00 | 12.80 | **27.20** | 12.69 | 7.52 | 35.86 | 21.61 |
| | **DISCO (10K)** | **11.40** | **20.20** | 4.66 | **28.00** | 31.58 | **1.79** | **13.65** | **14.80** | 21.60 | **14.32** | 6.86 | **40.56** | 36.43 |
| **Qwen3-VL-8B** | None | 40.80 | 75.20 | 42.00 | 44.33 | 72.75 | 32.79 | 40.25 | 43.20 | 76.80 | 50.00 | 42.74 | 66.61 | 93.28 |
| | Textual (10K) | 41.00 | **79.60** | 40.40 | 52.12 | 76.74 | **47.84** | 40.38 | 31.20 | 78.80 | 50.22 | 44.81 | 67.57 | 82.03 |
| | Textual-All (97K) | 37.70 | 75.80 | 41.42 | 50.20 | 71.66 | 23.03 | 45.50 | **50.40** | 71.60 | 59.54 | 50.07 | 62.29 | 86.05 |
| | DISCO *w/o $T_L$* (10K) | 44.30 | 77.60 | 43.22 | 55.32 | 76.23 | 44.39 | 43.39 | 44.40 | 76.80 | 51.84 | 48.67 | 70.59 | 88.62 |
| | **DISCO (10K)** | 42.90 | 75.90 | 55.95 | 56.11 | 80.50 | 33.91 | 56.77 | 44.40 | 78.40 | 65.51 | 59.12 | 71.48 | 84.44 |
| **Qwen3-VL-4B** | None | 28.70 | 68.90 | 24.47 | 29.71 | 32.00 | 14.14 | 12.19 | 42.00 | 66.80 | 14.43 | 28.70 | 28.97 | 0.00 |
| | Textual (10K) | **36.60** | 73.50 | 37.61 | 42.22 | 70.17 | 13.32 | 32.04 | **44.40** | 69.60 | 45.66 | 43.74 | 33.65 | **81.46** |
| | **DISCO (10K)** | 21.70 | **84.40** | 52.09 | 54.88 | 61.42 | 22.22 | 38.30 | 20.00 | 76.40 | 46.85 | 60.05 | 51.03 | 68.46 |
| **Qwen3-VL-32B** | None | 42.00 | 86.80 | 55.19 | 55.84 | 85.40 | 57.61 | 54.09 | 55.60 | 83.60 | 61.61 | 65.91 | 69.29 | 83.53 |
| | Textual (10K) | 49.60 | 89.80 | 64.38 | 61.87 | 89.64 | 65.78 | 63.19 | 54.00 | 82.80 | 63.99 | 70.71 | 66.11 | 84.16 |
| | **DISCO (10K)** | **64.20** | **93.50** | **72.04** | **64.75** | **89.85** | **68.40** | **65.47** | **66.80** | **86.80** | **68.11** | **74.10** | **71.70** | **88.40** |

*Table 2.* **Ablation study with Qwen3-VL-8B**, where *Full* indicates the combination of both DISCO and Table-GLS.

| Methods | HiTab | AIT-QA$_O$ | InfoTabs | PubHealthTab$_O$ |
|---|---|---|---|---|
| **Full** | 27.35 | **76.71** | 72.67 | **77.14** |
| *- GSE* | 24.30 | 62.82 | 72.09 | 74.92 |
| *- SSE* | 31.41 | 73.39 | 70.20 | 73.94 |
| *only* **Table-GLS** | 29.76 | 55.58 | **73.59** | 72.76 |
| **CoT** | 28.17 | 56.75 | 67.98 | 57.52 |
| **DISCO+CoT** | 26.40 | 73.78 | 71.00 | 68.33 |
| **RoT** | **33.88** | 55.58 | 61.26 | 58.29 |
| **DISCO+RoT** | 26.27 | 69.08 | 66.98 | 72.14 |

disentangled structure–content alignment. Meanwhile, DISCO exhibits strong data efficiency: with only 5K images, it already surpasses textual alignment with 10K. Performance improves with more data, peaking at 10K and remaining stable at 15K and 20K. This indicates that DISCO is not highly sensitive to data scale and can achieve strong table structure understanding with as few as 10K table images, while remaining robust as data increases.

**Table Reasoning.** Table 3 reports results on multimodal table reasoning tasks, covering question answering and fact verification under in-domain and out-of-domain settings. The main observations are: (I) Table-GLS outperforms direct answering on almost all benchmarks, highlighting the value of explicit reasoning strategies for table-centric reasoning. (II) After integrating DISCO, LVLMs show substantial gains in reasoning performance under direct answering settings, particularly for Qwen3-VL with relatively strong inherent table reasoning capabilities. This indicates that DISCO effectively enhances the inherent implicit structural modeling in LVLMs, enabling better learning of table

structure and content for reasoning. (III) Combining DISCO with Table-GLS achieves the best average score for both Gemma3n and Qwen3-VL, with notable gains on OOD benchmarks such as AIT-QA and PubHealthTab for Qwen3-VL, indicating the disentangled structure-content alignment provides more reliable evidence grounding, enabling the model to reason more accurately and robustly over unseen tables. (IV) Compared with heavily optimized LVLM, our inference framework achieves comparable performance with minimal supervision. Notably, when combining DISCO with Table-GLS, the model consistently matches or even surpasses optimized LVLM on several benchmarks, particularly on OOD benchmarks. This indicates that our framework effectively elicits the inherent table reasoning capability of LVLMs without relying on costly task-level fine-tuning. (V) As shown in Table 4, for larger-scale model, DiSCo and Table-GLS consistently improve performance on Qwen3-VL-32B, with the highest average accuracy of 71.84%, demonstrating that the structure-guided reasoning effectively enhances performance even with a larger model. Meanwhile, for closed-source model, Table-GLS substantially outperforms the vanilla GPT-5.4-mini, with average accuracy increasing from 56.64% to 66.16%, showing that our approach is robust across different LLMs, including open-source or closed-source models.

### 5.2. Ablation Study

We then examine the contribution of each alignment component in DISCO. As shown in Table 1, compared with the Textual baseline, DISCO without local content alignment (i.e., *w/o $T_L$*) already achieves consistent improvements,

*Table 3.* **Results on multimodal table reasoning tasks**, where # $TI_R$ indicates the number of table images used for training and $_O$ denotes the out-of-domain dataset that is unseen for all tested methods. The best result among the same category of methods are **bolded**.

| Models | Method | # $TI_R$ | Question Answering | | | | | Fact Verification | | | Avg. |
|---|---|---|---|---|---|---|---|---|---|---|---|
| | | | WTQ | HiTab | TAT-QA | AIT-QA$_O$ | TabMCQ$_O$ | TabFact | InfoTabs | PubHealthTab$_O$ | |
| *Closed-Source LVLM* | | | | | | | | | | | |
| GPT-4o-mini | DA | 0 | 30.06 | 20.94 | 28.37 | 54.01 | 56.07 | 46.54 | 59.28 | 48.61 | 42.99 |
| *Optimized LVLM* | | | | | | | | | | | |
| Table-LLaVA 7B | SFT | 82K | 18.43 | 10.09 | 12.82 | 5.48 | 44.51 | 59.85 | 65.26 | 51.03 | 33.43 |
| Table-LLaVA 13B | SFT | 82K | 20.41 | 10.85 | 15.67 | 6.06 | **51.51** | 65.00 | 66.91 | 48.46 | 35.61 |
| Qwen2-VL-7B-Table-R1 | GRPO | 20.6K | 50.30 | 58.20 | 48.06 | - | - | 73.40 | 62.80 | - | - |
| MiniCPM-V-2.6 8B | HIPPO | 55.2K | 55.77 | **63.00** | 60.75 | 66.14 | 39.65 | **82.27** | **75.74** | **73.32** | **64.58** |
| *Open-Source LVLM* | | | | | | | | | | | |
| Gemma3n-E4B | DA | 0 | 30.50 | 14.15 | 22.41 | 52.25 | **68.71** | 41.33 | **55.04** | **56.08** | 42.56 |
| Qwen3-VL-8B | DA | 0 | **49.47** | **35.47** | **36.01** | **71.04** | 50.53 | **70.21** | 45.59 | 44.18 | **50.31** |
| *Ours* | | | | | | | | | | | |
| Gemma3n-E4B-DɪSCᴏ | DA | 0 | 24.56 | 16.81 | 25.26 | **64.77** | **79.40** | 42.83 | 52.74 | 52.47 | 44.86 |
| | Table-GLS | 0 | **41.32** | **22.08** | **33.16** | 59.10 | 74.34 | **55.81** | **58.76** | **62.41** | **50.87** |
| Qwen3-VL-8B-DɪSCᴏ | DA | 0 | 50.16 | **33.50** | 37.95 | 75.73 | **56.37** | 69.28 | 63.44 | 63.34 | 56.22 |
| | Table-GLS | 0 | **57.11** | 27.35 | **40.54** | **76.71** | 48.88 | **75.41** | **72.67** | **77.14** | **59.48** |

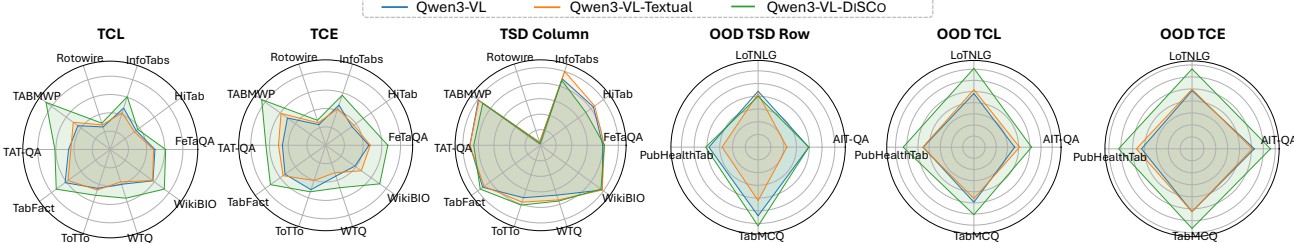

*Figure 3.* **Model performance on representative understanding tasks across various table layouts.**

*Table 4.* **The results on Qwen3-VL-32B and GPT-5.4-mini.** The best result are **bolded**.

| Methods | WTQ | AIT-QA$_O$ | InfoTabs | PubHealthTab$_O$ | Avg. |
|---|---|---|---|---|---|
| GPT-5.4-mini | 55.18 | 44.51 | 64.80 | 62.08 | 56.64 |
| +Table-GLS | **72.89** | **58.71** | **69.74** | **63.29** | **66.16** |
| Qwen3-VL-32B | 55.96 | 69.08 | 70.96 | 68.49 | 66.12 |
| +DɪSCᴏ | 57.00 | **77.87** | 72.20 | 69.57 | 69.16 |
| +DɪSCᴏ&Table-GLS | **65.08** | 73.19 | **75.28** | **73.79** | **71.84** |

indicating that disentangled structure alignment alone substantially enhances table layout understanding. Introducing local content alignment further boosts performance, particularly on structure-sensitive tasks such as TSD, TCL, and RCE, and improves robustness under out-of-domain settings. These results demonstrate that structure alignment and local content grounding are complementary and jointly essential for effective multimodal table understanding.

For Table-GLS, we first conduct ablations on reasoning task by removing the Global Structure Exploration (–*GSE*) or the Self-refined Sub-table Extraction (–*SSE*), with results reported in Table 2. Performance consistently drops when either stage is removed, demonstrating the importance of both global structure exploration and sub-table extraction for reliable table reasoning. Notably, eliminating *GSE* leads to substantial drops on all benchmarks, highlighting the

necessity of explicitly identifying task-relevant structural regions before reasoning. Interestingly, removing *SSE* slightly improves performance on HiTab, likely due to complex nested structures within the table of HiTab, where inaccurate sub-table extraction may introduce noisy evidence and hinder reasoning. We also conducted further experiments by directly applying Table-GLS to Qwen3-VL to evaluate the gains brought by DɪSCᴏ (*only* Table-GLS). It can be observed a significant performance decline on OOD benchmarks, fully demonstrating that our disentangled alignment enhances LVLMs' capabilities to understanding table images, enabling more robust and accurate reasoning on unseen tables. Finally, we further report the results with vanilla Chain-of-Thought (CoT) and Row-of-Thought (RoT) (Zhang et al., 2025b), a training-free, row-wise iterative reasoning framework for text TableQA. We can find that our DɪSCᴏ can also enhance the LVLM's ability to perform step-by-step reasoning by itself or row-wise iterative on table reasoning tasks, especially on OOD datasets such as AIT-QA and PubHealthTab. Meanwhile, compared with CoT and RoT, Table-GLS yields more stable and superior performance across most benchmarks, demonstrating the benefit of structure-guided global-to-local reasoning. We further analyze the token efficiency of various reasoning strategies in the Appendix G.

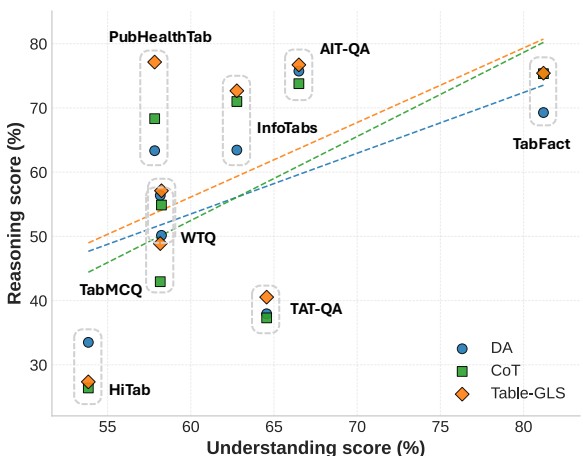

*Figure 4.* **Correlation between table understanding and reasoning performance of Qwen3-VL-DISCO.**

### 5.3. Impact of Table Layouts

To further evaluate the robustness of DISCO, we analyze the performance of Qwen3-VL-DISCO across table layouts of varying scale and structural complexity. As illustrated in Figure 3, DISCO consistently improves the performance of LVLM on most tasks across various table types, with the most significant gains observed in relatively small and compact tables, i.e., TABMWP and WikiBiO. In high-density tables such as ToTTo (average 35 rows) and Rotowire (average 33 rows and 19 columns), DISCO demonstrates superior robustness, effectively enhancing model performance across most tasks compared to text representation-based alignment methods. Crucially, the substantial margin maintained on OOD benchmarks, such as LoTNLG and PubHealthTab, further confirms that our method internalizes the universal logic of tabular organization rather than over-fitting to specific training distributions, thereby efficiently enhancing the LVLM's generalization to unseen layouts. More information about table layouts and results are presented in the Appendix F.

### 5.4. Understanding–Reasoning Correlation Analysis

Figure 4 depicts the relationship between table understanding accuracy and downstream reasoning performance on Qwen3-VL after applying DISCO alignment under different inference strategies. Overall, we observe a clear positive correlation that benchmarks with higher understanding scores consistently yield stronger reasoning performance, indicating that robust table understanding forms a critical foundation for reliable reasoning. Moreover, different reasoning paradigms exhibit distinct sensitivities to understanding quality. Direct Answering exhibits the weakest correlation, as it often bypasses explicit use of table structure and resorts to superficial matching, whereas CoT strengthens the consistency by introducing intermediate

*Table 5.* **Results on non-tabular tasks.** The best result are **bolded**.

| Models | ScienceQA | CRPE | HallusionBench | TextVQA |
|---|---|---|---|---|
| **MiniCPM-v2.6** | 95.19 | 76.32 | 63.83 | 75.54 |
| **+HIPPO** | **96.33** | **76.37** | 63.41 | **75.89** |
| **Qwen3-VL-8B** | 94.79 | 77.68 | 73.5 | 80.34 |
| +Textual (10K) | 94.94 | 77.85 | 72.24 | 79.91 |
| +DISCO (10K) | **95.09** | **77.92** | **74.97** | **80.83** |

*Table 6.* **Sensitivity analysis of Table-GLS to different prompt designs**, where $R$ denotes the rephrased prompt and $P$ denotes the prompt with prefix. The best result are **bolded**.

| Methods | WTQ | AIT-QA$_O$ | InfoTabs | PubHealthTab$_O$ | Avg. |
|---|---|---|---|---|---|
| **Qwen3-VL-8B** | 49.47 | 71.04 | 45.59 | 44.18 | 52.57 |
| +DISCO | 50.16 | 75.73 | 63.44 | 63.34 | 63.17 |
| +DISCO&Table-GLS | 57.11 | **76.71** | 72.67 | **77.14** | **70.91** |
| +DISCO&Table-GLS-R | 56.51 | 74.76 | 72.30 | 76.00 | 69.89 |
| +DISCO&Table-GLS-P | **57.48** | 74.17 | **72.74** | 76.67 | 70.27 |

reasoning steps that better exploit the LVLM's structure understanding. Notably, Table-GLS consistently achieves higher reasoning scores at comparable understanding levels, demonstrating that explicitly leveraging structure signals enables the LVLM to more effectively translate understanding gains into reasoning improvements.

### 5.5. Evaluation on General Reasoning Tasks

To examine the impact of table alignment strategies on general multimodal capabilities, we further evaluate DISCO on several non-tabular benchmarks: ScienceQA (Lu et al., 2022), hallucination-oriented datasets (CRPE (Wang et al., 2024) and HallusionBench (Guan et al., 2024)), and the OCR-based TextVQA (Singh et al., 2019). As shown in Table 5, alignment with textual representation exhibits performance degradation on some tasks compared to the original model, indicating that directly aligning table images with full textual representations may introduce alignment bias and over-specialization to linearized table formats, which can negatively affect general visual–language understanding. In contrast, applying DISCO consistently improves performance across all evaluated tasks. And compared to HIPPO, which is optimized based on extensive reasoning data, our DISCO achieves greater improvements in base model. This suggests that disentangling structure abstraction from semantic grounding helps the LVLM form explicit and reusable internal representations of structure and content, thereby eliciting its intrinsic reasoning capability rather than relying on superficial pattern matching. Consequently, the learned inductive biases transfer effectively beyond table-specific contexts, leading to more robust and generalizable multimodal reasoning.

### 5.6. Prompt Sensitivity of Table-GLS

To evaluate prompt sensitivity, we conduct additional experiments with two alternative prompt designs: **(1)**

*Table 7.* **Analysis of revise mechanism on Qwen3-VL-8B**. The best result are **bolded**.

| Methods | WTQ | AIT-QA$_O$ | InfoTabs | PubHealthTab$_O$ | Avg. |
|---|---|---|---|---|---|
| DiSCo&Table-GLS | **57.11** | **76.71** | **72.67** | **77.14** | **70.91** |
| - Revise | 55.78 | 70.84 | 72.37 | 75.23 | 68.56 |
| + Hal Check | 56.65 | 76.52 | 72.26 | 75.54 | 70.24 |

**Rephrase (R)**, a semantically equivalent rewrite of the original prompt, and **(2) Prefix (P)**, the original prompt augmented with a role-playing prefix. As shown in Table 6, the reasoning performance remains consistently strong across all variants, with average accuracy varying only minimally, demonstrating that Table-GLS is robust to reasonable prompt modifications. Thus, the improvements of our method primarily stem from its global-to-local structure-guided reasoning and disentangled structure-content alignment, rather than overfitting to a specific prompt template. Details are available in Appendix H.

### 5.7. Analysis of Revise Mechanism

We conduct an further ablation by removing the revision step ("- Revise") within *SSE* stage, where the LVLMs directly extracts the sub-table without self-refinement. As shown in Table 7, performance consistently drops, demonstrating its importance and effectiveness. To assess the reliability of the revision step, we randomly sample 50 instances and ask 4 undergraduates to judge whether LLM correctly identify and revise the errors in the initially generated reasoning plan. Finally, they find that 78%, 80%, 76%, and 72% of the cases are correctly revised, respectively. This indicates that the LLM can self-correct its reasoning plan with a relatively high success rate, supporting the reliability and effectiveness of the self-refinement mechanism.

We further incorporate an evidence checking step ("+ Hal Check") in *SSE* stage, requiring the LVLM to explicitly verify the existence of selected rows and columns before sub-table extraction. To our surprise, as shown in Table 7, explicit verification does not lead to performance gains; instead, our lightweight self-refinement proves simple and effective for this task.

### 5.8. Inference Costs

We finally measure the per-sample inference time and accuracy on three datasets provided by Fu et al. (2025). As shown in Table 8, while HIPPO achieves low latency, it lacks explicit structured reasoning and thus yields lower accuracy. For Tool-Augmented Reasoning method, although REFOCUS achieves comparable inference time on the simpler VWTQ_syn and strong accuracy, it still exhibits high latency due to the additional table editing steps. Meanwhile, it relies on auxiliary bounding box information of table regions to guide visual operations, which is difficult to obtain in

*Table 8.* **Comparison of inference time (s) and accuracy (%)**, where both ReFocus and DiSCo&Table-GLS are implemented based on Qwen3-VL-8B. All the methods are executed using the Transformers library under the same inference setting.

| Methods | VWTQ | | VWTQ_syn | | VTabFact | |
|---|---|---|---|---|---|---|
| | Time | Acc. | Time | Acc. | Time | Acc. |
| HIPPO | 0.72 | 55.87 | 0.75 | 58.00 | 0.86 | 86.00 |
| REFOCUS | 23.96 | **68.00** | 17.11 | **70.80** | 20.70 | 88.40 |
| DiSCo&Table-GLS | 14.89 | 62.27 | 15.19 | 68.40 | 12.11 | **91.20** |

real-world scenarios. In contrast, our DiSCo&Table-GLS achieves a better balance, requiring moderate inference time while maintaining competitive or superior accuracy (e.g., 91.20% on VTabFact), demonstrating a more favorable trade-off between efficiency and effectiveness.

## 6. Case Study

To further demonstrate the effectiveness of our framework, we present a case from WTQ in Figure 5. When using vanilla CoT, Qwen3-VL correctly identifies the relevant row but fails to ground the reasoning in accurate content evidence, leading to an incorrect arithmetic operation and a wrong final answer. This highlights the LVLM's limited ability to reliably exploit table content without structure-aware guidance. In contrast, applying Table-GLS directly enforces a structured reasoning process, enabling the model to successfully localizes the relevant row and result cell. However, without enhanced table representations, the extracted evidence is still imperfectly grounded, resulting in a erroneous reasoning. Finally, combining DiSCo with Table-GLS yields correct and interpretable reasoning. DiSCo enables the LVLM to more accurately understand table structure and content through disentangled alignment, allowing Table-GLS to precisely identify task-relevant rows and columns, extract clean sub-table evidence, and perform evidence-grounded reasoning. This example validate how disentangled structure–content alignment and global-to-local reasoning jointly enhance robustness and interpretability in multimodal table reasoning.

## 7. Conclusion

In this work, we introduce an efficient framework for multimodal table understanding and reasoning that enhances LVLMs without relying on large-scale annotations or external tools. Through disentangled structure-content alignment and lightweight structure-guided global-to-local reasoning, our framework effectively transfers LVLMs' intrinsic reasoning ability to tables. Extensive experiments demonstrate strong performance and robustness across diverse table tasks, highlighting a scalable and data-efficient solution for multimodal table reasoning.

## Impact Statement

This paper presents work whose goal is to advance the research on multimodal table reasoning with large vision-language models. There are many potential societal consequences of our work, none of which we feel must be specifically highlighted here.

## Acknowledgments

This work was supported in part by the National Science and Technology Major Program (2024ZD01NL00101), in part by the Science Fund for Creative Research Groups of the National Natural Science Foundation of China under Grant 62521006, in part by the National Natural Science Foundation of China (62406091, 62276077, U23B2055, U24A20328, 62350710797, 62506182), in part by the Xinjiang Science and Technology Development Plan for the Two Innovation Demonstration Zones along the Silk Road Economic Belt under Grant 2024LQ03003, in part by Guangdong S&T Program (2024B0101050003), in part by the Guangdong Basic and Applied Basic Research Foundation (2026A1515011718, 2024A1515011205), in part by Shenzhen Science and Technology Program (KQTD20240729102154066) and in part by the Major Key Project of PCL (PCL2025A03).

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

## Limitations

While our framework achieves strong performance with high scalability and data efficiency, it still has several limitations.

- First, DISCO still relies on the availability of the table structure–content decoupled data, and its effectiveness may vary when table layouts are extremely irregular or implicitly encoded.

- Second, although Table-GLS avoids external tools, its multi-stage reasoning introduces additional token generation compared to direct answering, which may increase inference cost in latency-sensitive settings.

- Finally, our study focuses on table-centric reasoning tasks; extending the proposed framework to more general structured visual reasoning scenarios remains an open direction for future work.

## A. Case Study

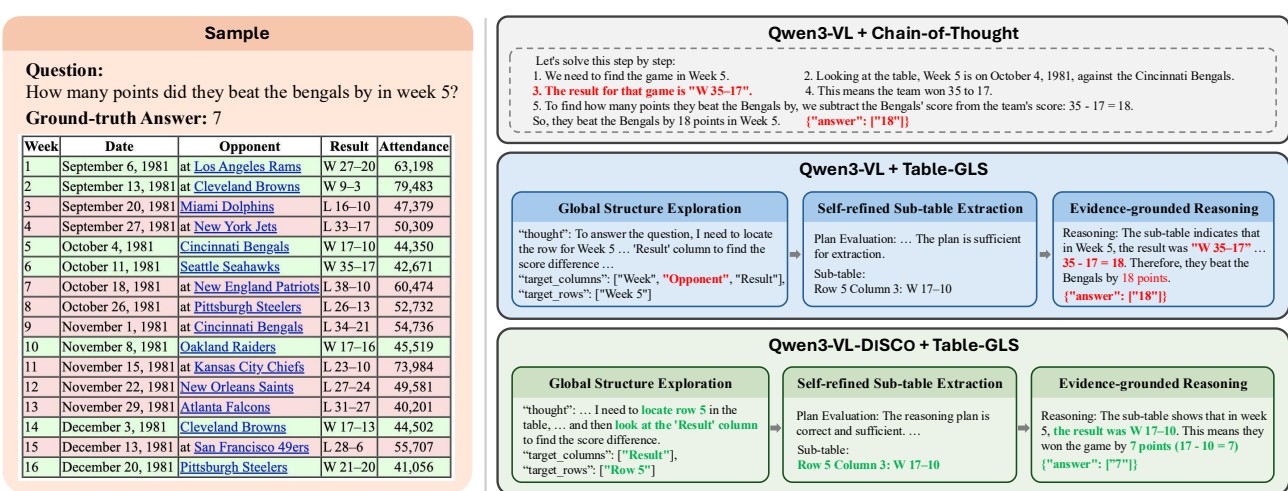

*Figure 5.* **An example of multimodal table reasoning task.**

## B. Data Contruction for DISCO

We first randomly sample 10K table images from various datasets within the pre-training corpus released by Zheng et al. (2024). For each image, we construct paired structure-alignment and content-alignment instances following the proposed DISCO framework, yielding a total of 30K image-text pairs for training.

**Structure Alignment.** We first replace the content information in the HTML, Markdown, or LaTeX representations corresponding to the table images with `[table content]`, retaining only the structural information. Then, we append "**Replace all the table contents with '[table content]', keeping the table structure intact.**" to the original queries provided by Zheng et al. (2024). The final data for structure alignment is constructed by combining the new query with its corresponding anonymized structure representation.

**Global Content Alignment.** We generated natural language descriptions for each table image, including the total number of rows and columns as well as the content of each cell, formatted as follows:

> *The table has m rows and n columns*
> *Row 1 Column 1: cell content*
> *Row 1 Column 2: cell content*
> *......*
> *Row m Column n: cell content*

To increase the diversity of samples, we design various instructions with similar semantics for global content alignment.

- <image>\nDescribe the table shown in the image in the following format.\nThe table has [m] rows and [n] columns.\nRow 1 Column 1: [Content]\nRow 1 Column 2: [Content]\n...\nRow m Column n: [Content]\n

- <image>\nDescribe the structure and content of the table in the image, listing each cell's information in the specified format.\nThe table has [m] rows and [n] columns.\nRow 1 Column 1: [Content]\nRow 1 Column 2: [Content]\n...\nRow m Column n: [Content]\n

- <image>\nProvide a thorough description of the table depicted in the image, including its dimensions and the content of each cell, following the format below.\nThe table has [m] rows and [n] columns.\nRow 1 Column 1: [Content]\nRow 1 Column 2: [Content]\n...\nRow m Column n: [Content]\n

- <image>\nExamine the table in the image and produce a comprehensive description that includes the number of rows and columns, as well as the content of each cell, formatted as shown.\nThe table has [m] rows and [n] columns.\nRow 1 Column 1: [Content]\nRow 1 Column 2: [Content]\n...\nRow m Column n: [Content]\n

- <image>\nTransform the table shown in the image into a detailed textual format, specifying the number of rows and columns, along with the content of each cell as illustrated below.\nThe table has [m] rows and [n] columns.\nRow 1 Column 1: [Content]\nRow 1 Column 2: [Content]\n...\nRow m Column n: [Content]\n

- <image>\nConvert the table displayed in the image into a detailed text description, adhering to the format provided below.\nThe table has [m] rows and [n] columns.\nRow 1 Column 1: [Content]\nRow 1 Column 2: [Content]\n...\nRow m Column n: [Content]\n

- <image>\nGenerate a structured textual representation of the table in the image, detailing each cell's content in the specified format.\nThe table has [m] rows and [n] columns.\nRow 1 Column 1: [Content]\nRow 1 Column 2: [Content]\n...\nRow m Column n: [Content]\n

- <image>\nAnalyze the table in the image and output a detailed textual description listing every cell in the following format.\nThe table has [m] rows and [n] columns.\nRow 1 Column 1: [Content]\nRow 1 Column 2: [Content]\n...\nRow m Column n: [Content]\n

- <image>\nRead the table content from the image and reconstruct its structure in text form as shown below.\nThe table has [m] rows and [n] columns.\nRow 1 Column 1: [Content]\nRow 1 Column 2: [Content]\n...\nRow m Column n: [Content]\n

- <image>\nProvide a detailed description of the table in the image, including the number of rows and columns, as well as the content of each cell, following the format below.\nThe table has [m] rows and [n] columns.\nRow 1 Column 1: [Content]\nRow 1 Column 2: [Content]\n...\nRow m Column n: [Content]\n

**Local Content Alignment.** We randomly select the cell at row $m$, column $n$ in the table and ask the model for its content. The corresponding label is "*Row m Column n: cell content*". We also generate 10 semantically similar instructions to enhance diversity.

- What is the exact value located at Row {R} and Column {C}?

- Retrieve the content of the cell at coordinate Row {R}, Column {C}.

- Perform a lookup for the data point at index Row {R}, Column {C}.

- Identify the specific data found in cell Row {R}, Column {C}.

- State the information present at Row index {R} and Column index {C}.

- Read the exact data from the cell defined by Row {R} and Column {C}.

- Query the table for the value at the coordinate (Row {R}, Column {C}).

- In the grid, what is present at the intersection of Row {R} and Column {C}?

- Return the single data point located at Row {R}, Column {C}.

- Content of the cell with indices Row {R}, Column {C}.

## C. Prompts for Table-GLS

---

**Global Structure Exploration $I_{GSE}$**

```
You are given a table image and a question.
Your task is to analyze the layout and headers of the table to locate the information
needed to answer the given question.

Please output in the following JSON format:
{{
    "thought": "Briefly explain your reasoning on which columns/rows are needed.",
    "target_columns": ["List the exact column headers required"],
    "target_rows": ["List the target row labels required"] or "Describe the condition
    to filter rows (e.g., 'Year is 2023 or 2024')",
}}

Question:
{question}
```

---

**Self-refined Sub-table Extraction $I_{SSE}$**

```
You are given a table image, a question and a reasoning plan with target rows and
columns.
First, evaluate whether the given reasoning plan is correct and sufficient for
answering the question. If the plan is incorrect or incomplete, revise it to obtain
a correct reasoning plan.
Then, based on the correct reasoning plan, extract the sub-table that is necessary to
answer the question.

Output strictly in the following format:
Plan Evaluation: "brief explanation of your judgment"
Sub-table:
Row m Column n: [Content]
...

Reasoning Plan:
{reasoning_plan}

Question:
{question}
```

---

Evidence-grounded Reasoning. $I_{EGR}$

```
You are given a table image, a question and a sub-table.
First, let's think step by step based on the given information.
Then provide the final concise answer in the JSON format {{\"answer\": \"<YOUR
ANSWER>\"}}.

Output in the following format:
Reasoning: "think step by step to answer the question"
{{\"answer\": \"<YOUR ANSWER>\"}}

Sub-table:
{subtable}

Question:
{question}
```

## D. Datasets

The statitiscs of table type used for DISCO alignment are shown in Table 9.

*Table 9.* **The statistics of DISCO alignment data**, where *# Tables* and *# Samples* indicated the number of table images and samples, respectively.

| Benchmarks | TABMWP | WTQ | FeTaQA | HiTab | TAT-QA | TabFact | InfoTabs | ToTTo | Rotowire | WikiBIO |
|---|---|---|---|---|---|---|---|---|---|---|
| **# Tables** | 2623 | 101 | 539 | 397 | 360 | 1694 | 123 | 2780 | 671 | 346 |
| **# Samples** | 8088 | 303 | 1617 | 1191 | 1134 | 5415 | 369 | 8691 | 2154 | 1038 |

The statistics of evaluation data for multimodal table understanding are shown in Table 10.

*Table 10.* **The statistics of multimodal table understanding data**, where *# Tables* and *# Samples* indicated the number of table images and samples, respectively.

| Task | TSD | TCL | RCE | MCD | TCE | OOD_TSD | OOD_TCL | OOD_RCE | OOD_TCE |
|---|---|---|---|---|---|---|---|---|---|
| **# Tables** | 1000 | 1000 | 1000 | 1000 | 1000 | 250 | 250 | 250 | 250 |
| **# Samples** | 1000 | 1000 | 1000 | 1000 | 1000 | 250 | 250 | 250 | 250 |

The statistics of evaluation data for multimodal table reasoning are shown in Table 11.

*Table 11.* **The statistics of multimodal table reasoning data**, where *# Tables* and *# Samples* indicated the number of table images and samples, respectively.

| Benchmark | WTQ | HiTab | TAT-QA | TabFact | InfoTabs | TabMCQ | AIT-QA | PubHealthTab |
|---|---|---|---|---|---|---|---|---|
| **# Tables** | 421 | 535 | 231 | 1045 | 600 | 50 | 111 | 362 |
| **# Samples** | 4344 | 1576 | 772 | 6845 | 5400 | 1029 | 511 | 1942 |

## E. Implementation Details

The hyperparameters for DISCO alignment training are shown in Table 12.

*Table 12.* **Hyperparameters for DISCO alignment training.**

| Models | Lora_rank | Lora_alpha | Global Batch Size | Learning rate | Epoch |
|---|---|---|---|---|---|
| **Gemma3-12B** | 8 | 16 | 64 | 1e-4 | 1 |
| **LLaVA-v1.6-7B** | 8 | 16 | 64 | 1e-4 | 1 |
| **Gemma3n-E4B** | 8 | 16 | 64 | 1e-5 | 1 |
| **Qwen3-VL-8B** | 8 | 16 | 64 | 4e-5 | 1 |

For multimodal table understanding tasks, we directly use the query provided by MMTab for testing. For table reaoning tasks, we append "*Provide the answer in the JSON format {"answer": "<YOUR ANSWER>"} directly without any other explanation.*" and "*Think step by step and output the final answer in the JSON format {"answer": "<YOUR ANSWER>"}*" after the original question from each benchmark (for TQA tasks) or the query provided by MMTab (for TFV tasks) for direct answering and chain-of thought, respectively.

Following Zheng et al. (2024), we use accuracy for the evaluation of table question answering and fact verification tasks. For TSD, we compute accuracy for the predicted row count and column count separately. For TCE and TCL, we calculate the accuracy at cell level. For MCD, we apply the cell-level F1 score. And for RCE, we compute the cell-level F1 score separately for the extracted rows and columns. All the experiments are finished on 4 NVIDIA H20 GPUs with 96GB memory.

# F. Impact of Table Layouts

We first analyze the scale of tables associated with each benchmark in MMTab, and then manually assign a complexity level (high/medium/low) to each table based on its structure, such as scale, header nesting, and cell-merging characteristics. The results are shown in Table 13.

*Table 13.* **The statistics of the scale and complexity of tables in each benchmark.**

| Benchmarks | LoTNLG | TABMWP | HiTab | TabFact | TAT-QA | Rotowire | InfoTabs | AIT-QA | PubHealthTab | TabMCQ | FeTaQA | ToTTo | HiTab_t2t | WikiBIO | WTQ |
|---|---|---|---|---|---|---|---|---|---|---|---|---|---|---|---|
| **Avg. Rows** | 14.49 | 6.45 | 19.37 | 14.07 | 9.73 | 32.63 | 11.21 | 13.39 | 9.24 | 13.62 | 15.16 | 34.97 | 22.38 | 9.70 | 27.78 |
| **Avg. Cols** | 6.18 | 2.19 | 9.84 | 6.24 | 3.92 | 19.00 | 2.04 | 5.50 | 3.87 | 4.10 | 5.70 | 6.71 | 8.16 | 3.02 | 6.33 |
| **Complexity** | Medium | Low | High | Medium | Low | High | Low | High | Low | Low | Medium | High | High | Low | High |

Figure 6 illustrates additional experimental results on multimodal table understanding performance across various table layouts. Overall, the observed trends are consistent with those in the main paper, showing that DISCO brings stable improvements across different table layouts, with more pronounced gains on unseen tables.

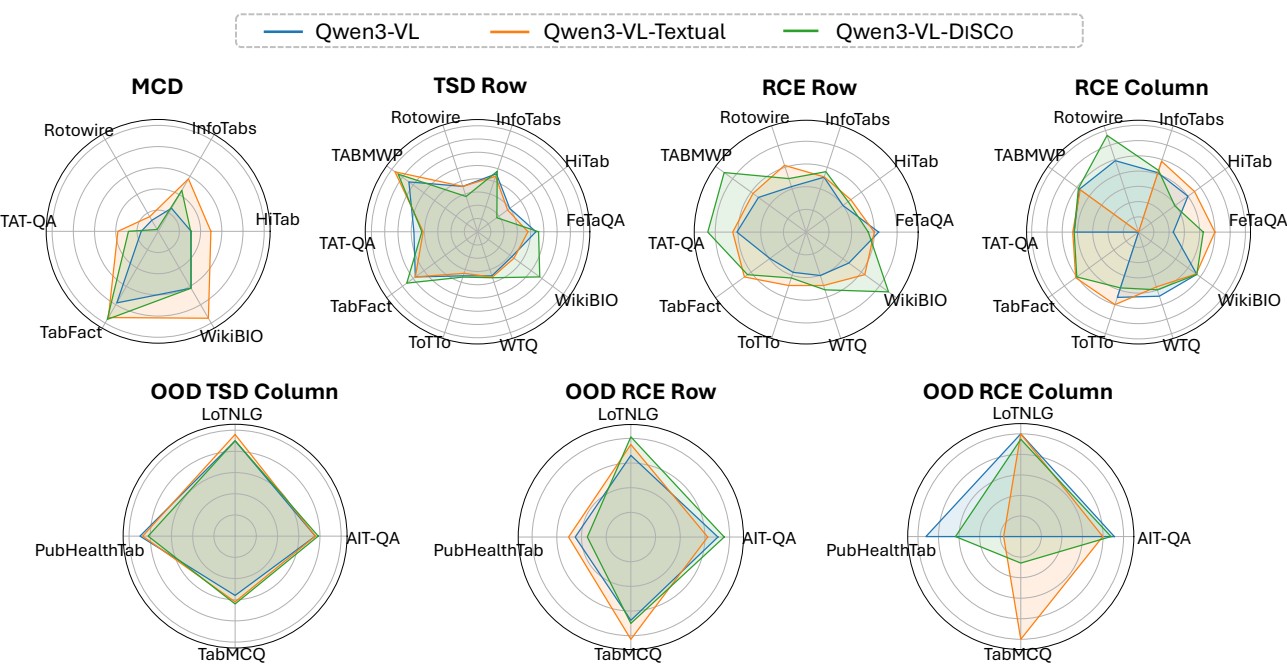

*Figure 6.* **Model performance on representative understanding tasks across various table layouts.**

*Table 14.* **Complete results on multimodal table understanding tasks.** *None* denotes the original model, *Textual* and *Textual-All* denote the multimodal alignment with textual representations, i.e., HTML, Markdown and LaTeX, where the former includes **the same table image** as in our DISCO and the latter encompasses all alignment data provided by MMTab. *w/o $T_L$* indicates the removal of the local content alignment.

| Models | Alignment | TSD | | TCL | RCE | | MCD | TCE | OOD TSD | | OOD TCE | OOD TCL | OOD RCE | |
|---|---|---|---|---|---|---|---|---|---|---|---|---|---|---|
| | | Row | Column | | Row | Column | | | Row | Column | | | Row | Column |
| TableLlama+Oracle | None | 5.30 | 4.40 | 0.82 | 4.34 | 5.26 | - | 9.35 | - | - | - | - | - | - |
| TableLlama+OCR | None | 3.90 | 3.70 | 0.65 | 2.82 | 2.39 | - | 3.95 | - | - | - | - | - | - |
| Table-LLaVA 7B | Textual-All | 33.10 | **33.20** | 29.31 | **31.43** | 37.93 | **17.14** | 19.45 | 25.20 | **16.40** | 11.28 | 26.10 | **21.97** | 18.14 |
| Table-LLaVA 13B | Textual-All | **34.40** | 27.60 | **29.68** | 31.07 | **41.49** | 16.52 | **19.53** | **31.60** | 14.80 | **11.38** | **26.17** | 21.94 | **18.67** |
| Gemma3-12B | None | 12.40 | 33.10 | 3.01 | 25.46 | 47.65 | 2.41 | 9.26 | 15.20 | 40.80 | 8.46 | 2.86 | 21.62 | 56.95 |
| | Textual | 12.00 | 30.00 | 4.33 | 26.32 | 46.94 | 0.52 | 9.34 | 18.80 | 34.40 | 10.41 | 6.46 | 32.35 | 63.89 |
| | DISCO *w/o $T_L$* | **21.30** | **37.90** | 15.50 | 28.79 | 46.22 | 2.45 | 17.90 | 25.20 | **45.20** | 22.78 | 21.70 | 33.10 | 64.23 |
| | DISCO | 20.30 | 33.00 | 17.25 | 43.65 | 54.47 | 2.84 | 21.53 | 28.80 | 34.80 | 26.14 | 20.34 | 42.31 | 71.39 |
| LLaVA-v1.6-7B | None | 2.50 | 5.40 | 1.05 | 10.91 | 10.72 | **0.51** | 3.52 | 3.60 | 6.00 | 3.47 | 1.26 | 14.25 | 9.27 |
| | Textual | 3.40 | 10.70 | **4.10** | 17.07 | 10.82 | 0.27 | 2.65 | 2.40 | 8.40 | 2.82 | 2.53 | 22.60 | 11.34 |
| | DISCO *w/o $T_L$* | 5.20 | 10.80 | 1.70 | 16.43 | **14.95** | 0.00 | 5.07 | 7.60 | 6.40 | 6.07 | 1.80 | **33.02** | 13.33 |
| | DISCO | **12.20** | **20.20** | 2.84 | **18.76** | 8.30 | 0.02 | **16.28** | **13.20** | **15.60** | **18.98** | **2.73** | 28.00 | 9.35 |
| Gemma3n-E4B | None | 5.50 | 15.00 | **8.50** | 24.82 | **32.44** | 1.11 | 9.02 | 6.80 | 18.40 | 9.00 | 10.99 | 19.38 | 29.77 |
| | Textual | 9.50 | 19.70 | 8.30 | 24.54 | 23.25 | 0.71 | 8.64 | 10.80 | 23.60 | 10.20 | **12.32** | 24.20 | **37.06** |
| | DISCO *w/o $T_L$* | 9.90 | 20.20 | 4.71 | 27.45 | 23.69 | 0.76 | 11.00 | 12.80 | **27.20** | 12.69 | 7.52 | 35.86 | 21.61 |
| | DISCO | **11.40** | **20.20** | 4.66 | **28.00** | 31.58 | **1.79** | **13.65** | **14.80** | 21.60 | **14.32** | 6.86 | **40.56** | 36.43 |
| Qwen3-VL-8B | None | 40.80 | 75.20 | 42.00 | 44.33 | 72.75 | 32.79 | 40.25 | 43.20 | 76.80 | 50.00 | 42.74 | 66.61 | 93.28 |
| | Textual (10K) | 41.00 | **79.60** | 40.40 | 52.12 | 76.74 | **47.84** | 40.38 | 31.20 | 78.80 | 50.22 | 44.81 | 67.57 | 82.03 |
| | Textual-All (97K) | 37.70 | 75.80 | 41.42 | 50.20 | 71.66 | 23.03 | 45.50 | **50.40** | 71.60 | 59.54 | 50.07 | 62.29 | 86.05 |
| | DISCO *w/o $T_L$* (10K) | 44.30 | 77.60 | 43.22 | 55.32 | 76.23 | 44.39 | 43.39 | 44.40 | 76.80 | 51.84 | 48.67 | 70.59 | 88.62 |
| | DISCO (5K) | 40.80 | 76.70 | 40.56 | 55.38 | 83.30 | 45.17 | 46.51 | 43.20 | 84.00 | 53.25 | 45.74 | 56.83 | 78.31 |
| | DISCO (10K) | 42.90 | 75.90 | **55.95** | 56.11 | 80.50 | 33.91 | **56.77** | 44.40 | 78.40 | **65.51** | **59.12** | 71.48 | 84.44 |
| | DISCO (15K) | **45.40** | 77.10 | 42.67 | **61.78** | **86.47** | 45.09 | 49.86 | 46.40 | **79.60** | 54.56 | 46.21 | 71.75 | 87.13 |
| | DISCO (20K) | 43.50 | 74.80 | 41.97 | 59.95 | 82.60 | 44.07 | 49.30 | 46.40 | 78.80 | 55.10 | 46.27 | **83.04** | **98.59** |
| Qwen3-VL-4B | None | 28.70 | 68.90 | 24.47 | 29.71 | 32.00 | 14.14 | 12.19 | 42.00 | 66.80 | 14.43 | 28.70 | 28.97 | 0.00 |
| | Textual | **36.60** | 73.50 | 37.61 | 42.22 | 70.17 | 13.32 | 32.04 | **44.40** | 69.60 | 45.66 | 43.74 | 33.65 | **81.46** |
| | DISCO | 21.70 | **84.40** | 52.09 | 54.88 | 61.42 | 22.22 | 38.30 | 20.00 | 76.40 | **46.85** | **60.05** | 51.03 | 68.46 |
| Qwen3-VL-32B | None | 42.00 | 86.80 | 55.19 | 55.84 | 85.40 | 57.61 | 54.09 | 55.60 | 83.60 | 61.61 | 65.91 | 69.29 | 83.53 |
| | Textual | 49.60 | 89.80 | 64.38 | 61.87 | 89.64 | 65.78 | 63.19 | 54.00 | 82.80 | 63.99 | 70.71 | 66.11 | 84.16 |
| | DISCO | **64.20** | **93.50** | **72.04** | **64.75** | **89.85** | **68.40** | **65.47** | **66.80** | **86.80** | **68.11** | **74.10** | **71.70** | **88.40** |

## G. Token Efficient Analysis

Table 15 reports the average number of tokens generated during inference under different reasoning strategies. Overall, CoT-based method produces relatively short reasoning traces, while Table-GLS introduces longer outputs due to its explicit multi-stage global-to-local reasoning process. Notably, incorporating DISCO consistently reduces the token length for vanilla CoT across all benchmarks, indicating that structure–content disentangled alignment enables more concise and focused reasoning. For Table-GLS, DISCO slightly increases token usage on some benchmarks, reflecting richer and more explicit structural exploration and sub-table verification. Notably, this increase is not uniform and remains controlled across tasks. Therefore, these results suggest that DISCO improves reasoning efficiency by reducing redundant generation in free-form reasoning, while complementing Table-GLS with more structured and interpretable reasoning traces rather than indiscriminately increasing verbosity.

*Table 15.* **The statistics of the scale and complexity of tables in each benchmark.**

| Methods | WTQ | HiTab | TAT-QA | TabFact | InfoTabs | TabMCQ | AIT-QA | PubHealthTab | Avg |
|---|---|---|---|---|---|---|---|---|---|
| CoT | 337.73 | 281.50 | 221.75 | 317.63 | 191.75 | 261.40 | 219.25 | 181.92 | 251.62 |
| DISCO+CoT | 248.59 | 230.63 | 194.38 | 230.79 | 116.49 | 156.34 | 215.53 | 128.42 | 190.15 |
| Table-GLS | 719.73 | 424.44 | 421.70 | 559.16 | 356.61 | 426.27 | 341.41 | 385.60 | 454.37 |
| DISCO+Table-GLS | 727.58 | 539.53 | 373.85 | 696.35 | 539.63 | 377.33 | 355.46 | 493.41 | 512.89 |

## H. Prompt Sensitivity of Table-GLS

For **Prefix (P)** variant, we prepend a role-playing prefix to the original prompts to examine the sensitivity to stylistic guidance. Specifically, in the Global Structure Exploration stage, we add *"You are a careful table structure analyst."*; in the Self-refined Sub-table Extraction stage, we use *"You are a precise table reasoning evaluator."*; and in the Evidence-grounded Reasoning stage, we include *"You are a meticulous table reasoning assistant."*

For the Rephrase variant, we rewrite the prompts at each stage while preserving their original semantics and functionality. The rephrased prompts for each stage are as follows:

---

**Rephrased Prompt of Global Structure Exploration**

```
You are provided with a table image along with a question.
Your goal is to examine the table's structure and headers to identify where
the relevant information is located for answering the question.

Please respond using the following JSON format:
{{
    "thought": "Briefly explain your reasoning on which columns/rows are needed.",
    "target_columns": ["List the exact column headers required"],
    "target_rows": ["List the target row labels required"] or "Describe the condition
    to filter rows (e.g., 'Year is 2023 or 2024')",
}}

Question:
{question}
```

---

**Rephrased Prompt of Self-refined Sub-table Extraction**

```
You are provided with a table image, a question, and a reasoning plan that specifies
target rows and columns.
First, assess whether the given reasoning plan is accurate and sufficient to answer the
question. If it is incorrect or incomplete, modify it to form a valid reasoning plan.
Then, using the corrected reasoning plan, extract the sub-table required to answer
the question.

Output strictly in the following format:
Plan Evaluation: "brief explanation of your assessment"
Sub-table:
Row m Column n: [Content]
...

Reasoning Plan:
{reasoning_plan}

Question:
{question}
```

---

Rephrased Prompt of Evidence-grounded Reasoning

```
You are provided with a table image, a question, and a sub-table.
First, reason through the problem step by step using the given information.
Then, return the final concise answer in JSON format {{\"answer\": \"<YOUR ANSWER>\"}}.

Output in the following format:
Reasoning: "think step by step to answer the question"
{{\"answer\": \"<YOUR ANSWER>\"}}

Sub-table:
{subtable}

Question:
{question}
```

