# OpenReview forum: "Decoupling Skeleton and Flesh: Efficient Multimodal Table Reasoning with Disentangled Alignment and Structure-aware Guidance"
_ICML.cc/2026/Conference — ICML 2026 spotlight_

### Official Review · Reviewer_hKF2 · 2026-03-07

**Soundness:** 3
**Presentation:** 3
**Significance:** 3
**Originality:** 3
**Overall Recommendation:** 4
**Confidence:** 3

**Summary:**

This paper addresses the challenges of multimodal table reasoning in Large Vision-Language Models by introducing two main contributions. First, it proposes the DiSCo training framework, which decouples the alignment of table structural abstraction (skeleton) and semantic content extraction (flesh) during training. This approach enhances the model's fundamental table understanding capabilities with relatively low training overhead and data requirements. Second, building upon this, the authors introduce Table-GLS, a global-to-local structure-aware reasoning framework. By guiding the LVLM through structured exploration and targeted subtable extraction, Table-GLS enables reliable, end-to-end table reasoning, mitigating hallucinations and avoiding the cascading errors common in traditional tool-augmented pipelines.

**Compliance With Llm Reviewing Policy:**

Affirmed.

**Final Justification:**

My concerns have been partially addressed and I would like to keep my positive score.

**Key Questions For Authors:**

1. Baseline Comparisons: The proposed DiSCo and Table-GLS modules are very interesting. To further highlight their specific strengths, it would be highly beneficial to see a comparison with more recent TableQA strategies. Could the authors consider adding a controlled comparison ?
2. Efficiency and Latency: A core motivation of this work is to address the "high latency and complexity" associated with traditional tool-augmented methods. However, it seems there isn't a direct empirical comparison of latency between the proposed method and a representative tool-augmented baseline. Could the authors consider providing a brief latency comparison to further strengthen this important claim?
3. Details on the Revise Mechanism: I found the "Revise" step in the subtable extraction process to be a promising direction. Could the authors kindly provide a bit more detail on the criteria the LVLM uses to self-determine if a revision is needed? I have a minor concern regarding how the framework prevents the model from becoming overconfident in its own hallucinations without an external verifier. Clarifying this mechanism would be much appreciated.

**Limitations:**

Yes.

**Strengths And Weaknesses:**

### Strengths

- Originality and Data Efficiency: The proposed "Decoupled Skeleton and Content (DiSCo)" training paradigm offers a novel perspective by disentangling complex table understanding into structural abstraction and content extraction. This design significantly reduces the reliance on massive amounts of layout-specific training data, demonstrating exceptional data efficiency.
- Architectural Robustness: The method achieves pure end-to-end vision-language reasoning, completely eliminating the dependence on external OCR or parsing tools. This fundamentally avoids the cascading errors typically caused by parser failures or layout misidentifications, thereby enhancing the model's robustness.
- Interpretability and Hallucination Mitigation: The global-to-local subtable extraction strategy in the Table-GLS framework not only aligns well with human cognitive logic when reading tables, but also effectively truncates the context window during the final reasoning stage by filtering out redundant information. This mechanism naturally mitigates the hallucinations commonly seen in LVLMs.

### Weaknesses

- Lack of Depth in the Self-Refinement Mechanism: The "Revise" step within the subtable extraction process appears somewhat superficial. It lacks objective evaluation criteria and robust multi-turn verification designs. Without the assistance of external verifiers, relying solely on the LVLM's self-introspection makes the model highly susceptible to overconfidence, making substantial error correction difficult to achieve.
- Insufficient and Outdated Baseline Comparisons: In the ablation studies, comparing DiSCo + Table-GLS primarily against the standard Vanilla CoT is insufficient, as Vanilla CoT is notoriously poor at handling table topologies. To more rigorously validate the effectiveness of the two proposed modules, stronger and more recent baselines (e.g., modern TableQA strategies like RoT) should be introduced for controlled variable comparisons. On one hand, comparing "Standard LVLM + Modern TableQA strategy" against "DiSCo + the same strategy" would explicitly prove the fundamental alignment gains brought by DiSCo. On the other hand, Table-GLS should be horizontally compared with these advanced reasoning strategies (e.g., comparing "DiSCo + Modern TableQA strategy" vs. "DiSCo + Table-GLS") to comprehensively demonstrate the superiority of the Table-GLS reasoning mechanism itself.
- Claims of "High Latency/Inefficiency" Lack Direct Empirical Support: The authors repeatedly emphasize in the motivation that existing tool-augmented methods suffer from "high latency and complexity." However, the experimental section (including Appendix G) only analyzes token generation counts and provides no comparative data on actual end-to-end wall-clock time. This omission leaves the core claim of being "more efficient than external tool pipelines" without direct empirical validation.

---

> ### Author Rebuttal · Authors · 2026-03-31
>
> Thanks for your insightful comments.
> >**W1:** Details and Analysis of Revise Mechanism.
>
> **R1:** We clarify both its design principle and effectiveness below.
>
> **(1) Design principle and preventing overconfidence.**
> Our goal is to maintain efficiency without introducing external tools.
> To this end, we ask the LLM to self-evaluate the reasoning plan by re-grounding it in the original table image and question. Specifically, the LLM explicitly checks whether the selected rows/columns are sufficient and relevant to answer the question based on the table image, and revise them if necessary. The prompt is shown in Appendix C.
>
> **(2) Effectiveness of the Revise step.**
> We conduct an ablation by removing the revision step ("- Revise"), where the LLM directly extracts the sub-table without self-refinement. As shown in Table 1, performance consistently drops, demonstrating its importance and effectiveness.
>
> To assess the reliability of the revision step, we randomly sample 50 instances and ask 4 undergraduates to judge whether LLM correctly identify and revise the errors in the initially generated reasoning plan. Finally, they find that 78%, 80%, 76%, and 72% of the cases are correctly revised, respectively. This indicates that the LLM can self-correct its reasoning plan with a relatively high success rate, supporting the reliability and effectiveness of the self-refinement mechanism.
>
> **(3) Incorporating explicit hallucination checking.**
> Motivated by your comment, we further an evidence checking step ("+ Hal Check") in stage 2, requiring the LLM to explicitly verify the existence of selected rows and columns before sub-table extraction. To our surprise, as shown in Table 1, explicit verification does not lead to performance gains; instead, our lightweight self-refinement proves simple and effective for this task. Exploring efficient and effective verification mechanisms remains an important direction for future work.
>
> Table 1: Ablation study on 4 representative datasets.
> |Methods|WTQ|AIT-QA|InfoTabs|PubHealthTab|Avg.|
> |:-|:-:|:-:|:-:|:-:|:-:|
> |DiSCo+Table-GLS|**57.11**|**76.71**|**72.67**|**77.14**|**70.91**|
> |- Revise|55.78|70.84|72.37|75.23|68.56|
> |+ Hal Check|56.65|76.52|72.26|75.54|70.24|
> >**W2:** Insufficient and Outdated Baseline Comparisons.
>
> **R2:** We incorporate RoT as a representative modern TableQA strategy and conduct controlled comparisons.
>
> First, comparing RoT with DiSCo+RoT, we observe consistent improvements across most datasets, demonstrating that DiSCo provides complementary alignment gains even when combined with another table reasoning strategy.
>
> Second, comparing DiSCo+RoT with DiSCo+Table-GLS, our method achieves clear and consistent gains, outperforming RoT across all benchmarks, especially on out-of-domain datasets such as AIT-QA and PubHealthTab. This validates the superiority of Table-GLS over existing table reasoning strategies.
>
> Together, these results show that DiSCo and Table-GLS are both effective and complementary, where DiSCo improves the underlying alignment, and Table-GLS provides a stronger reasoning strategy beyond existing methods.
>
> Table 2: Comparison with RoT. All the methods are implemented based on Qwen3-VL-8B.
> |Methods|WTQ|AIT-QA|InfoTabs|PubHealthTab|Avg.|
> |:-|:-:|:-:|:-:|:-:|:-:|
> |Vanilla|49.47|71.04|45.59|44.18|52.57|
> |RoT|**64.02**|55.58|61.26|58.29|59.79|
> |DiSCo+RoT|55.36|69.08|66.98|72.14|65.89|
> |DiSCo+Table-GLS|57.11|**76.71**|**72.67**|**77.14**|**70.91**|
> >**W3:** Claims of "High Latency/Inefficiency" Lack Direct Empirical Support.
>
> **R3:** To further support our claim on efficiency, we conducted additional experiments measuring the end-to-end inference time and accuracy for each type of methods. Implementation details are provided in R3 within the response to Reviewer QwtF.
>
> As shown in Table 3, although ReFocus (a tool-augmented reasoning method) performs relatively faster inference on simpler VWTQ_syn, it still incurs notably high latency on other datasets, due to additional table editing steps. Meanwhile, it relies on **auxiliary bounding box information** for table regions, which is hard to obtain in real-world scenarios. Whereas DiSCo+Table-GLS reduces inference time to 12–15s, achieving up to 41% speedup. Meanwhile, our method maintains competitive or superior accuracy. Besides, although HIPPO (SFT/RL-based fine-tuning method) achieves low latency, it lacks structured reasoning and yields lower performance.
>
> These results provide direct empirical evidence that, compared to tool-augmented reasoning, our method achieves lower latency while maintaining strong reasoning performance, validating its efficiency advantage in practical scenarios.
>
> Table 3: Inference time (seconds) / accuracy (%) across datasets.
> |Methods|VWTQ|VWTQ_syn|VTabFact|
> |:-|:-:|:-:|:-:|
> |HIPPO|0.72/55.87|0.75/58.00|0.86/86.00|
> |ReFocus|23.96/68.00|17.11/70.80|20.7/88.40|
> |DiSCo+Table-GLS|14.89/62.27|15.19/68.40|12.11/91.20|
>
> We will add all above results and analysis to the revised paper.

---

> > ### Author Rebuttal · Reviewer_hKF2 · 2026-04-03
> >
> > Thank you for the detailed response. However, given the limited depth of methodological innovation in the overall work, I have decided to maintain my current score.

---

> > > ### Author Response · Authors · 2026-04-03
> > >
> > > Thank you for your thoughtful follow-up and for confirming that our responses have addressed your concerns.
> > >
> > > We are especially grateful for your recognition in the following aspects:
> > >
> > > - **Originality and data efficiency of the DiSCo paradigm**: our decoupled skeleton–content design provides a novel perspective on table understanding and significantly reduces the reliance on large-scale training data, demonstrating strong data efficiency.
> > > - **Architectural robustness of the end-to-end framework**: our method eliminates the need for external OCR or parsing tools, thereby avoiding cascading errors and improving the robustness of table reasoning.
> > > - **Interpretability and hallucination mitigation of Table-GLS**: our global-to-local reasoning strategy aligns well with human cognitive processes and effectively reduces hallucinations by filtering redundant context.
> > >
> > > Regarding the concern on the depth of methodological innovation, we would like to further clarify that our work goes beyond introducing new components, and instead proposes a unified paradigm for multimodal table understanding and reasoning. We summarize below the key novelties and advantages of our proposed paradigm:
> > > 1. **Disentangled alignment for transferable capability**: unlike conventional multimodal alignment strategies that align table images with serialized textual representations, our paradigm explicitly decouples structure and content. This allows LVLMs to transfer their inherent ability to table images in a data-efficient manner and learn consistent with the their intrinsic generation and understanding patterns, improving generalization and stability.
> > > 2. **High data efficiency**: our framework does not rely on large-scale annotated data and achieves strong performance with limited supervision, significantly reducing the cost of data collection and improving practicality in low-resource settings.
> > > 3. **Efficiency–accuracy balance**: our method achieves competitive or superior accuracy while substantially reducing end-to-end latency, providing a better trade-off compared to prior approaches that typically sacrifice efficiency for performance.
> > >
> > > We believe these combined benefits demonstrate the practical impact and novelty of our approach, particularly for real-world table reasoning scenarios.
> > >
> > > As mentioned in our rebuttal, we will incorporate all additional experiments and analyses (including stronger baseline comparisons, detailed evaluation of the revise mechanism, and latency measurements) into the revised version to further strengthen the paper.
> > >
> > > Thank you again for your valuable feedback and support.

---

### Official Review · Reviewer_13xk · 2026-03-11

**Soundness:** 3
**Presentation:** 3
**Significance:** 3
**Originality:** 3
**Overall Recommendation:** 5
**Confidence:** 3

**Summary:**

Compared with existing approaches that either directly train models on table annotation data or rely on external tools to assist table understanding, this paper proposes a new paradigm for table data. Through a decoupled training design, the proposed paradigm enables efficient alignment and understanding of tables with only a small amount of data. Experimental results further demonstrate the effectiveness of the method.

**Compliance With Llm Reviewing Policy:**

Affirmed.

**Final Justification:**

I have reviewed the authors’ rebuttal and decided to keep my current score unchanged.

**Key Questions For Authors:**

Please provide a response discussing the weaknesses mentioned above.

**Limitations:**

yes

**Strengths And Weaknesses:**

**Strengths:**
1. The paper proposes a more efficient and generalizable structured alignment format for tables. Compared with the diverse and fragmented formatting schemes used in existing work, enabling the model to learn a representation that is more consistent with its intrinsic generation and understanding patterns is clearly meaningful and potentially valuable.
2. The experimental design is fairly comprehensive, and the method is validated on multiple backbone models, which demonstrates its effectiveness and a certain degree of generalizability.

**Weaknesses:**
1. The paper uses a training set of 10k examples, but it does not show how performance changes as the data scale varies. As a result, it is still difficult to fully assess the method’s sensitivity to data size and its data efficiency.

---

> ### Author Rebuttal · Authors · 2026-03-31
>
> We appreciate both your recognition of our work and your insightful review. We address each of your concerns below.
>
> >**W1:** The paper uses a training set of 10k examples, but it does not show how performance changes as the data scale varies. As a result, it is still difficult to fully assess the method’s sensitivity to data size and its data efficiency.
>
> **R1:** To further highlight the data scalability and efficiency of our method, we conducted additional experiments varying the alignment data size from 5K to 20K, comparing DiSCo against traditional HTML/Markdown/LaTeX-based textual alignment.
>
> As shown in Table 1, DiSCo consistently outperforms textual alignment across all data scales, demonstrating the advantage of our disentangled structure–content Alignment. Meanwhile, DiSCo exhibits strong data efficiency: with only 5K samples, it already surpasses textual alignment with 10K data. Performance improves with more data, peaking at 61.95 (10K) and remaining stable at 61.07 (15K) and 61.87 (20K). This indicates that DiSCo is not highly sensitive to data scale and can achieve strong table structure understanding with as few as 10K table images, while remaining robust as data increases.
>
> Overall, these results show that our method achieves high data efficiency and stable scaling behavior, validating its effectiveness even in relatively low-resource settings.
>
> Table 1: Effect of alignment data scale (TI$_A$) on table understanding performance.
>
> | Alignment | TI$_A$ | TSD_Row | TSD_Column | TCL | RCE_Row | RCE_Column | MCD | TCE | OOD_TSD_Row | OOD_TSD_Column | OOD_TCE | OOD_TCL | OOD_RCE_Row | OOD_RCE_Column | Avg. |
> |:---|:---:|:---:|:---:|:---:|:---:|:---:|:---:|:---:|:---:|:---:|:---:|:---:|:---:|:---:|:---:|
> | None | 0 | 40.80 | 75.20 | 42.00 | 44.33 | 72.75 | 32.79 | 40.25 | 43.20 | 76.80 | 50.00 | 42.74 | 66.61 | 93.28 | 55.44 |
> | Textual | 10K | 41.00 | **79.60** | 40.40 | 52.12 | 76.74 | 47.84 | 40.38 | 31.20 | 78.80 | 50.22 | 44.81 | 67.57 | 82.03 | 56.36 |
> | Textual | 97K | 37.70 | 75.80 | 41.42 | 50.20 | 71.66 | 23.03 | 45.50 | **50.40** | 71.60 | 59.54 | 50.07 | 62.29 | 86.05 | 55.78 |
> | DiSCo | 5K | 40.80 | 76.70 | 40.56 | 55.38 | 83.30 | **45.17** | 46.51 | 43.20 | **84.00** | 53.25 | 45.74 | 56.83 | 78.31 | 57.67 |
> | DiSCo | 10K | 42.90 | 75.90 | **55.95** | 56.11 | 80.50 | 33.91 | **56.77** | 44.40 | 78.40 | **65.51** | **59.12** | 71.48 | 84.44 | **61.95** |
> | DiSCo | 15K | **45.40** | 77.10 | 42.67 | **61.78** | **86.47** | 45.09 | 49.86 | 46.40 | 79.60 | 54.56 | 46.21 | 71.75 | 87.13 | 61.07 |
> | DiSCo | 20K | 43.50 | 74.80 | 41.97 | 59.95 | 82.60 | 44.07 | 49.30 | 46.40 | 78.80 | 55.10 | 46.27 | **83.04** | **98.59** | 61.87 |
>
> We will add all above results and analysis to the revised paper. Thanks for your comments.

---

> > ### Author Rebuttal · Reviewer_13xk · 2026-04-03
> >
> > Thank you to the authors for the detailed response. I have no further questions.

---

> > > ### Author Response · Authors · 2026-04-03
> > >
> > > Thank you for your kind follow-up and for confirming that your concerns have been fully addressed.
> > >
> > > We sincerely appreciate your recognition of our work, particularly in the following aspects:
> > >
> > > - **Structured and generalizable alignment format:** our proposed alignment paradigm is more efficient and unified compared to the diverse and fragmented formatting schemes in prior work.
> > > - **Alignment with model intrinsic capabilities**: our design enables the model to learn representations that are more consistent with its intrinsic generation and understanding patterns.
> > > - **Comprehensive experimental validation and generalizability**: our experimental design is fairly comprehensive and that validation across multiple backbone models effectively demonstrates both the method’s effectiveness and a certain degree of generalizability.
> > >
> > > We are especially encouraged by your acknowledgement of the **data efficiency** and **generalizability** of our method, which are central goals of this work.
> > >
> > > We will incorporate the additional experiments on data scaling, along with the corresponding analysis, into the revised version of the paper to further strengthen its completeness and clarity.
> > >
> > > Thank you again for your valuable feedback and support.

---

### Official Review · Reviewer_QwtF · 2026-03-14

**Soundness:** 2
**Presentation:** 3
**Significance:** 2
**Originality:** 3
**Overall Recommendation:** 4
**Confidence:** 3

**Summary:**

The paper focuses on the efficiency and scalability of table reasoning. Specifically, the authors introduce DISCO, a disentagled structure-content alignment framework that enhances MLLMs' abilities on understanding table structure and content, and Table-GLS, a global-to-lcoal structure-aware inference framework without extensive training or external tools. The reported results show that the proposed DISCO method improves understanding performance efficiently with only 10K alignment images. Additionally, combining DISCO with Table-GLS achieves the best average score for both Gemma3n-E4B and Qwen3-VL-8B, with notable gains on OOD benchmarks, such as AIT-QA and PubHealthTab.

**Compliance With Llm Reviewing Policy:**

Affirmed.

**Final Justification:**

The authors provided further experimental results to address my concerns, so I increase my score to 4.

**Key Questions For Authors:**

Could the authors provide the analysis of inference costs for each involved model, such as SFT/RL-based Fine-tuning, Tool-Augmented Reasoning, and DISCO + Table-GLS.

**Limitations:**

Yes.

**Strengths And Weaknesses:**

__Strengths__:

1. The proposed two methods effectively improve the reasoning abilities of VLMs while maintaining efficiency, as shown in the results of Table 1 and 2, which may be useful for the field of table reasoning in LLMs.

2. The motivation and presentation of this paper is good. For example, the Figure 1 clearly indicates the difference between proposed methods and previous ones.

3. The results on table understanding and reasoning tasks verify the effectiveness of proposed methods. This decomposition is necessary to follow and understand.

__Weaknesses__:

1. Even though the resuls of Table 2 are promising, the proposed method Table-GLS is prompt-driven. Importantly, the paper do not provide any analysis of the sensitivity or robustness of used prompting templates.

2. The table understanding tasks include comparisons with 32B model while such comparisons are missing in the table reasoning tasks. This omission might weaken the verification of the effectiveness of proposed method. I am curious about the model performance if using in-house LLMs like GPT-5 or Gemini in the prompting stage.

---

> ### Author Rebuttal · Authors · 2026-03-31
>
> Thank you for the insightful comments, our point-to-point responses to your comments are given below.
> >**W1:** Even though the resuls of Table 2 are promising, the proposed method Table-GLS is prompt-driven. Importantly, the paper do not provide any analysis of the sensitivity or robustness of used prompting templates.
>
> **R1:** To evaluate prompt sensitivity, we conducted additional experiments on four representative reasoning datasets from MMTab using two alternative prompt designs: (1) **Rephrase (R)**, a semantically equivalent rewrite of the original prompt, and (2) **Prefix (P)**, the original prompt augmented with a role-playing prefix. As shown in Table 1, the reasoning performance remains consistently strong across all variants, with average accuracy varying only minimally (<1.1%), demonstrating that Table-GLS is robust to reasonable prompt modifications. These results indicate that the improvements of our method primarily stem from its **global-to-local structure-guided reasoning** and **disentangled structure-content alignment**, rather than overfitting to a specific prompt template.
>
> Table 1: Sensitivity analysis of Table-GLS to different prompt designs.  *R*: the rephrased prompt; *P*: the prompt with prefix. All experiments are conducted using Qwen3-VL-8B.
> |Methods|WTQ|AIT-QA|InfoTabs|PubHealthTab|Avg.|
> |:-|:-:|:-:|:-:|:-:|:-:|
> |Vanilla|49.47|71.04|45.59|44.18|52.57|
> |DiSCo|50.16|75.73|63.44|63.34|63.17|
> |DiSCo+Table-GLS|57.11|76.71|72.67|77.14|70.91|
> |DiSCo+Table-GLS-R|56.51|74.76|72.30|76.00|69.89|
> |DiSCo+Table-GLS-P|57.48|74.17|72.74|76.67|70.27|
> >**W2:** The table understanding tasks include comparisons with 32B model while such comparisons are missing in the table reasoning tasks. This omission might weaken the verification of the effectiveness of proposed method. I am curious about the model performance if using in-house LLMs like GPT-5 or Gemini in the prompting stage.
>
> **R2:** To verify the effectiveness with larger and stronger models, we conducted additional experiments using both Qwen3-VL-32B and the latest GPT-5.4-mini.
>
> As shown in Table 2, on Qwen3-VL-32B, DiSCo and Table-GLS consistently outperform the vanilla model. Specifically, Table-GLS achieves the highest average accuracy of 71.84%, demonstrating that the structure-guided reasoning effectively enhances performance even with a larger model.
>
> On GPT-5.4-mini, as shown in Table 3, Table-GLS substantially outperforms the vanilla model, with average accuracy increasing from 56.64% to 66.16%, showing that our approach is robust across different LLMs, including open-source or in-house models.
>
> Together, these results validate that DiSCo and Table-GLS are model-agnostic and robust across different LLM backbones and scales.
>
> Table 2: The results on Qwen3-VL-32B.
> |Methods|WTQ|AIT-QA|InfoTabs|PubHealthTab|Avg.|
> |:-|:-:|:-:|:-:|:-:|:-:|
> |Vanilla|55.96|69.08|70.96|68.49|66.12|
> |DiSCo|57.00|**77.87**|72.20|69.57|69.16|
> |DiSCo+Table-GLS|**65.08**|73.19|**75.28**|**73.79**|**71.84**|
>
> Table 3: The results on GPT-5.4-mini.
> |Methods|WTQ|AIT-QA|InfoTabs|PubHealthTab|Avg.|
> |:-|:-:|:-:|:-:|:-:|:-:|
> |Vanilla|55.18|44.51|64.80|62.08|56.64|
> |Table-GLS|**72.89**|**58.71**|**69.74**|**63.29**|**66.16**|
> >**Q1:** Could the authors provide the analysis of inference costs for each involved model, such as SFT/RL-based Fine-tuning, Tool-Augmented Reasoning, and DISCO + Table-GLS.
>
> **R3:** We measure the per-sample reasoning time and accuracy for each type of methods on the three datasets used in the ReFocus paper, i.e. VWTQ (with 750 samples), VWTQ_syn (with 250 samples) and VTabFact (with 250 samples). For a fair comparison, both ReFocus and DiSCo+Table-GLS are implemented based on Qwen3-VL-8B, and all methods are executed using the Transformers library under the same inference setting.
>
> As shown in Table 4, while HIPPO (SFT/RL-based fine-tuning method) achieves low latency, it lacks explicit structured reasoning and thus yields lower accuracy. For Tool-Augmented Reasoning method, although ReFocus achieves comparable inference time on the simpler VWTQ_syn and strong accuracy, it still exhibits high latency due to the additional table editing steps. Meanwhile, it relies on **auxiliary bounding box information** of table regions to guide visual operations, which is difficult to obtain in real-world scenarios.
> In contrast, DiSCo+Table-GLS achieves a better balance, requiring moderate inference time while maintaining competitive or superior accuracy (e.g., 91.20% on VTabFact), demonstrating a more favorable trade-off between efficiency and effectiveness.
>
> Table 4: Inference time (seconds) / accuracy (%) across datasets.
> |Methods|VWTQ|VWTQ_syn|VTabFact|
> |:-|:-:|:-:|:-:|
> |HIPPO|0.72 / 55.87|0.75 / 58.00|0.86 / 86.00|
> |ReFocus|23.96 / 68.00|17.11 / 70.80|20.7 / 88.40|
> |DiSCo+Table-GLS|14.89 / 62.27|15.19 / 68.40|12.11 / 91.20|
>
> Due to time and space limitations, we will report additional results and analysis in the revised paper.

---

> > ### Author Rebuttal · Reviewer_QwtF · 2026-04-04
> >
> > The authors have repondes all of my questions, so I increase my score to 4.

---

> > > ### Author Response · Authors · 2026-04-04
> > >
> > > Thank you for your thoughtful follow-up and for increasing your score. We sincerely appreciate your time and consideration in re-evaluating our work.
> > >
> > > We are particularly grateful for your recognition of the following aspects:
> > >
> > > - **Effectiveness with efficiency**: our methods improve table reasoning performance while maintaining efficiency, which aligns with our core goal of balancing effectiveness and scalability.
> > > - **Clear motivation and presentation**: the motivation is well-explained and Figure 1 clearly illustrates the differences between our approach and prior methods, which we carefully designed to enhance clarity.
> > > - **Strong empirical validation**: the experimental results on both table understanding and reasoning tasks effectively demonstrate the benefits of our decomposition into structure and content, supporting the design of our framework.
> > >
> > > As discussed in our rebuttal, we will incorporate all additional experiments and analyses (including prompt sensitivity, larger model evaluation, and inference cost comparisons) into the revised version to further strengthen the paper.
> > >
> > > Thank you again for your valuable feedback and support.

---

### Decision · Program_Chairs · 2026-04-30

**Decision:**

Accept (spotlight)

**Comment:**

The reviewers all agree to accept the paper, and the AC concurs.